# LAST-ITERATE CONVERGENCE PROPERTIES OF REGRET MATCHING ALGORITHMS IN GAMES*

**Yang Cai**
Yale
yang.cai@yale.edu

**Gabriele Farina**
MIT
gfarina@mit.edu

**Julien Grand-Clément**
HEC Paris
grand-clement@hec.fr

**Christian Kroer**
Columbia
ck2945@columbia.edu

**Chung-Wei Lee**
USC
leechung@usc.edu

**Haipeng Luo**
USC
haipengl@usc.edu

**Weiqiang Zheng**
Yale
weiqiang.zheng@yale.edu

## ABSTRACT

We study last-iterate convergence properties of algorithms for solving two-player zero-sum games based on Regret Matching$^+$ (RM$^+$). Despite their widespread use for solving real games, virtually nothing is known about their last-iterate convergence. A major obstacle to analyzing RM-type dynamics is that their regret operators lack Lipschitzness and (pseudo)monotonicity. We start by showing numerically that several variants used in practice, such as RM$^+$, predictive RM$^+$ and alternating RM$^+$, all lack last-iterate convergence guarantees even on a simple $3 \times 3$ matrix game. We then prove that recent variants of these algorithms based on a *smoothing* technique, *extragradient RM$^+$* and *smooth Predictive RM$^+$*, enjoy asymptotic last-iterate convergence (without a rate), $1/\sqrt{t}$ best-iterate convergence, and when combined with restarting, linear-rate last-iterate convergence. Our analysis builds on a new characterization of the geometric structure of the limit points of our algorithms, marking a significant departure from most of the literature on last-iterate convergence. We believe that our analysis may be of independent interest and offers a fresh perspective for studying last-iterate convergence in algorithms based on non-monotone operators.

## 1 INTRODUCTION

Saddle-point optimization problems have attracted significant research interest with applications in generative adversarial networks (Goodfellow et al., 2020), imaging (Chambolle and Pock, 2011), market equilibrium (Kroer et al., 2019), and game-solving (Von Stengel, 1996). Matrix games provide an elementary saddle-point optimization setting, where the set of decisions of each player is a simplex and the objective function is bilinear. Matrix games can be solved via self-play, where each player employs a *regret minimizer*, such as online gradient descent ascent (GDA), multiplicative weight updates (MWU), or Regret Matching$^+$ (RM$^+$). A well-known folk theorem shows that the *average* of the strategies visited at all iterations converges to a Nash equilibrium, at a rate of $O(1/\sqrt{T})$ for GDA, MWU and RM$^+$, and at a rate of $O(1/T)$ for predictive variants of GDA and MWU (Syrgkanis et al., 2015; Rakhlin and Sridharan, 2013).

In recent years, there has been increasing interest in the *last-iterate* convergence properties of algorithms for saddle-point problems (Daskalakis and Panageas, 2019; Golowich et al., 2020; Wei et al., 2021; Lee et al., 2021). There are multiple reasons for this. First, since no-regret learning is often viewed as a plausible method of real-world gameplay, it would be desirable to have the actual strategy iterates converge to an equilibrium, instead of only having the average converge. Secondly,

---

*Authors are listed in alphabetic order.

suppose self-play via no-regret learning is being used to compute an equilibrium. In that case, iterate averaging can often be cumbersome, especially when deep-learning components are involved in the learning approach, since it may not be possible to average the outputs of a neural network. Thirdly, iterate averaging may be slower to converge since the convergence rate is limited by the extent to which early "bad" iterates are discounted in the average. Even in the simple matrix game setting, interesting questions arise when considering the last-iterate convergence properties of widely used algorithms. For instance, both GDA and MWU may diverge (Bailey and Piliouras, 2018; Cheung and Piliouras, 2019), whereas their predictive counterparts, Optimistic GDA (OGDA) (Daskalakis et al., 2018; Mertikopoulos et al., 2019; Wei et al., 2021) and Optimistic MWU (OMWU) (Daskalakis and Panageas, 2019; Lei et al., 2021) converge at a linear rate under some assumptions on the matrix games. Furthermore, it has been demonstrated that OGDA and the Extragradient Algorithm (EG) have a last-iterate convergence rate of $O(1/\sqrt{T})$ without any assumptions on the game (Cai et al., 2022; Gorbunov et al., 2022b).

On the other hand, very little is known about the last-iterate convergence properties of RM$^+$ and its variants. This is in spite of the fact that RM$^+$-based algorithms have been used in *every* case of solving extremely large-scale poker games (Bowling et al., 2015; Moravčík et al., 2017; Brown and Sandholm, 2018), one of the primary real-world instantiations of using learning dynamics to solve huge-scale zero-sum games. Poker games are modeled as *extensive-form games* (EFGs), a class of sequential games that capture imperfect information and stochasticity. RM$^+$ is a regret minimizer for strategy sets that are probability simplexes, and thus is not directly applicable to EFGs. In order to apply it in EFGs, it is combined with *counterfactual regret minimization* (Zinkevich et al., 2007), which enables solving an EFG by instantiating simplex regret minimizers at every decision point. Despite its very strong empirical performance, the last-iterate behavior of any variant of RM$^+$ is still not understood. It has been found empirically that the last-iterate in RM$^+$ combined with CFR (in alternating self play) may achieve better performance than the average iterate (Bowling et al., 2015), but on the other hand the last iterate may diverge even on rock-paper-scissors (Lee et al., 2021). Moreover, unlike OGDA and OMWU, the predictive variants of RM$^+$ do not enjoy a theoretical speed up: Predictive RM$^+$ (PRM$^+$), a predictive variant of RM$^+$, was introduced in Farina et al. (2021) and shown to achieve very good empirical performance in some games. However, it still only guarantees $O(1/\sqrt{T})$ ergodic convergence for matrix games (unlike OGDA and OMWU). To address this, Farina et al. (2023) introduce variants of RM$^+$ with $O(1/T)$ ergodic convergence for matrix games, namely, Extragradient RM$^+$ (ExRM$^+$) and Smooth Predictive RM$^+$ (SPRM$^+$), and show that these algorithms also enjoy strong practical ergodic performance on matrix games.

Overall, while much is known about the last-iterate convergence properties of OGDA, OMWU, and EG, these algorithms are not widely used in real large-scale EFG applications, and very little is known about the last-iterate convergence of the RM$^+$-based algorithms that are used in practice. As a first step toward understanding the last-iterate convergence properties of RM$^+$-algorithms in EFGs, we study both theoretically and empirically the last-iterate behavior of RM$^+$ and its variants for the special case of matrix games. Our **main results** are as follows.

1. **Divergence of RM$^+$.** We provide numerical evidence that RM$^+$ and important variants of RM$^+$, including alternating RM$^+$ and PRM$^+$, may fail to have asymptotic last-iterate convergence and best-iterate convergence. Conversely, we also prove that RM$^+$ *does* have last-iterate convergence in a very restrictive setting, where the matrix game admits a strict Nash equilibrium (Theorem 1)[1].

2. **Asymptotic convergence under the Minty condition.** Zero-sum game solving with RM$^+$ can be seen as solving a variational inequality with a *non-monotone* operator, which we call the regret operator (see Equation (3)). This is a significant departure from the existing results, *e.g.*, for the extragradient algorithm and optimistic gradient on *monotone* games (Cai et al., 2022).

   **Technical Contribution:** We first show that ExRM$^+$ and SPRM$^+$ exhibit asymptotic last-iterate convergence in the duality gap, enabled by our observation that the regret operator satisfies the *Minty condition* (see (6)), a condition strictly weaker than (pseudo)monotonicity. However, convergence in the duality gap is weaker than convergence *in iterates*, the usual form of convergence guarantee for monotone operators. An important benefit of convergence *in iterates* is that it

---

[1] Meng et al. (2023) studies RM$^+$ in strongly-convex-strongly-concave games and proves its asymptotic convergence (without rate) to the unique equilibrium. Theorem 1's proof largely follows from (Meng et al., 2023) with the key observation that the properties of strict NE suffices to adapt the analysis in (Meng et al., 2023) to zero-sum games (not strongly-convex-strongly-concave games) with a strict NE.

ensures the stability of the dynamics, in the sense that the strategies asymptotically stop changing (see Section 4.1 for a detailed discussion). Although the Minty condition is a well-studied concept in variational inequalities (Kinderlehrer and Stampacchia, 2000), it does *not* generally imply last-iterate convergence in iterates. Nevertheless, we are able to show that both ExRM$^+$ and SPRM$^+$ exhibit asymptotic last-iterate convergence in iterates as well. Our proof relies on a new characterization of the geometric structure of the limit points of the learning dynamics, which may be of independent interest for analyzing last-iterate convergence in other algorithms.

3. **Finite-time convergence.** We then show a $O(1/\sqrt{t})$-rate for the duality gap for the *best* iterate after $t$ iterations of both algorithms. Building on this observation, we finally introduce new variants of ExRM$^+$ and SPRM$^+$ that restart whenever the distance between two consecutive iterates has been halved, and prove that they enjoy *linear last-iterate convergence*.

4. **Experiments.** We validate the last-iterate convergence of ExRM$^+$ and SPRM$^+$ (including their restarted variants that we propose) numerically on four instances of matrix games, including Kuhn poker and Goofspiel. We also note that while vanilla RM$^+$, alternating RM$^+$, and PRM$^+$ may not converge, alternating PRM$^+$ exhibits a surprisingly fast last-iterate convergence.

## 2 PRELIMINARIES ON REGRET MATCHING$^+$

**Notation.** We write $\mathbf{0}$ for the vector with $0$ on every component and $\mathbf{1}_d$ for the vector in $\mathbb{R}^d$ with $1$ on every component. We use the convention that $\mathbf{0}/0 = (1/d)\mathbf{1}_d$. $\Delta^d$ is the simplex: $\Delta^d = \{x \in \mathbb{R}_+^d \mid \langle x, \mathbf{1}_d \rangle = 1\}$. For $x \in \mathbb{R}$, we write $[x]^+$ for the positive part of $x$: $[x]^+ = \max\{0, x\}$, and we overload this notation to vectors component-wise. We use $\|\cdot\|$ to denote the $\ell_2$ norm.

In this paper, we study iterative algorithms for solving the following matrix game:

$$\min_{x \in \Delta^{d_1}} \max_{y \in \Delta^{d_2}} x^\top A y \tag{1}$$

for a *payoff matrix* $A \in \mathbb{R}^{d_1 \times d_2}$. We define $\mathcal{Z} = \Delta^{d_1} \times \Delta^{d_2}$ to be the set of feasible pairs of strategies. The duality gap of a pair of feasible strategy $(x, y) \in \mathcal{Z}$ is defined as $\mathrm{DualityGap}(x, y) := \max_{y' \in \Delta^{d_2}} x^\top A y' - \min_{x' \in \Delta^{d_1}} x'^\top A y$. Note that we always have $\mathrm{DualityGap}(x, y) \geq 0$, and it is well-known that $\mathrm{DualityGap}(x, y) \leq \epsilon$ implies that the pair $(x, y) \in \mathcal{Z}$ is an $\epsilon$-Nash equilibrium of the matrix game (1). When both players of (1) employ a *regret minimizer*, a well-known folk theorem shows that the averages of the iterates generated during self-play converge to a Nash equilibrium (NE) of the game (Freund and Schapire, 1999). This framework can be instantiated with any regret minimizers, for instance, online mirror descent, follow-the-regularized leader, regret matching, and optimistic variants of these algorithms. We refer to (Hazan et al., 2016) for an extensive review on regret minimization. From here on, we focus on solving (1) via Regret Matching$^+$ and its variants. To describe these algorithms, it is useful to define for a strategy $x \in \Delta^d$ and a loss vector $\ell \in \mathbb{R}^d$, the negative instantaneous regret vector $f(x, \ell) = \ell - x^\top \ell \cdot \mathbf{1}_d$,[2] and the normalization operator $g : \mathbb{R}_+^{d_1} \times \mathbb{R}_+^{d_2} \to \mathcal{Z}$ such that

$$g(z) = \left( \frac{z_1}{\|z_1\|_1}, \frac{z_2}{\|z_2\|_1} \right), \forall z = (z_1, z_2) \in \mathbb{R}_+^{d_1} \times \mathbb{R}_+^{d_2} \tag{2}$$

**Regret Matching$^+$ (RM$^+$) and its variants.** We describe Regret Matching$^+$ (RM$^+$) in Algorithm 1 (Tammelin, 2014).[3] It maintains two sequences: a sequence of joint *aggregate payoffs* $(R_x^t, R_y^t) \in \mathbb{R}_+^{d_1} \times \mathbb{R}_+^{d_2}$ updated using the instantaneous regret vector, and a sequence of joint strategies $(x^t, y^t) \in \mathcal{Z}$ directly normalized from the aggregate payoff. The update rules are stepsize-free and only perform closed-form operations (thresholding and rescaling). Predictive RM$^+$ (PRM$^+$, Algorithm 2) (Farina et al., 2021) incorporates *predictions* of the next losses faced by each player (using the most recent observed losses) when computing the strategies at each iteration, akin to predictive/optimistic online mirror descent (Rakhlin and Sridharan, 2013; Syrgkanis et al., 2015). We describe two popular variants of RM$^+$ and PRM$^+$, *Alternating RM$^+$* (Tammelin et al., 2015; Burch

---

[2]Here, $d$ can be either $d_1$ or $d_2$. That is, we overload the notation $f$ so its domain depends on the inputs.

[3]Typically, RM$^+$ and PRM$^+$ are introduced as regret minimizers that return a sequence of decisions against any sequence of losses (Tammelin et al., 2015; Farina et al., 2021). For conciseness, we directly present them as self-play algorithms for solving matrix games, as in Algorithm 1 and Algorithm 2.

et al., 2019) and *Alternating PRM$^+$*, in Appendix A. Using alternation, the updates between the two players are asynchronous, and at iteration $t$, the second player observes the choice $x^{t+1}$ of the first player when choosing their own decision $y^{t+1}$. Alternation leads to faster empirical performance for solving matrix and extensive-form games, even though the theoretical guarantees remain the same as for vanilla RM$^+$ (Burch et al., 2019; Grand-Clément and Kroer, 2023).

---

**Algorithm 1** Regret Matching$^+$ (RM$^+$)

1: **Initialize**: $(R_x^0, R_y^0) = \mathbf{0}$, $(x^0, y^0) \in \mathcal{Z}$
2: **for** $t = 0, 1, \dots$ **do**
3: $\quad R_x^{t+1} = [R_x^t - f(x^t, Ay^t)]^+$
4: $\quad R_y^{t+1} = [R_y^t + f(y^t, A^\top x^t)]^+$
5: $\quad (x^{t+1}, y^{t+1}) = g(R_x^{t+1}, R_y^{t+1})$

**Algorithm 2** Predictive RM$^+$ (PRM$^+$)

1: **Initialize**: $(R_x^0, R_y^0) = \mathbf{0}$, $(x_0, y_0) \in \mathcal{Z}$
2: **for** $t = 0, 1, \dots$ **do**
3: $\quad R_x^{t+1} = [R_x^t - f(x^t, Ay^t)]^+$
4: $\quad R_y^{t+1} = [R_y^t + f(y^t, A^\top x^t)]^+$
5: $\quad (x^{t+1}, y^{t+1}) = g([R_x^{t+1} - f(x^t, Ay^t)]^+, [R_y^{t+1} + f(y^t, A^\top x^t)]^+)$

---

Despite its strong empirical performance, it is unknown if alternating PRM$^+$ enjoys ergodic convergence. In contrast, based on the aforementioned folk theorem and the regret guarantees of RM$^+$ (Tammelin, 2014), alternating RM$^+$ (Burch et al., 2019), and PRM$^+$ (Farina et al., 2021), the duality gap of the average strategy of all these algorithms goes down at a rate of $O(1/\sqrt{T})$. However, we will show in the next section that the iterates $(x^t, y^t)$ themselves may not converge. We also note that despite the connections between PRM$^+$ and predictive online mirror descent, PRM$^+$ does not achieve $O(1/T)$ ergodic convergence, because of its lack of stability (Farina et al., 2023).

**Extragradient RM$^+$ and Smooth Predictive RM$^+$** We now describe two theoretically-faster variants of RM$^+$ recently introduced in Farina et al. (2023). To provide a concise formulation, we first need some additional notation. First, we define the clipped positive orthant $\Delta_\geq^{d_i} := \{u \in \mathbb{R}_+^{d_i} : u^\top \mathbf{1}_{d_i} \geq 1\}$ for $i = 1, 2$ and $\mathcal{Z}_\geq = \Delta_\geq^{d_1} \times \Delta_\geq^{d_2}$. For a point $z \in \mathcal{Z}_\geq$, we often write it as $z = (Rx, Qy)$ for positive real numbers $R$ and $Q$ such that $(x, y) = g(z)$. Moreover, we define the operator $F : \mathcal{Z}_\geq \to \mathbb{R}^{d_1 + d_2}$ as follows: for $z \in \mathcal{Z}_\geq$, let $(x, y) = g(z)$ and

$$F(z) = \begin{bmatrix} f(x, Ay) \\ f(y, -A^\top x) \end{bmatrix} = \begin{bmatrix} Ay - x^\top Ay \cdot \mathbf{1}_{d_1} \\ -A^\top x + x^\top Ay \cdot \mathbf{1}_{d_2} \end{bmatrix} \tag{3}$$

Note that $F$ is Lipschitz continuous over $\mathcal{Z}_\geq$ with $L_F = \sqrt{6} \|A\|_{op} \max\{d_1, d_2\}$ (Farina et al., 2023), but $F$ is *not* Lipschitz continuous over the larger set $\mathbb{R}_+^{d_1 + d_2}$, as we show in Appendix B. This is because in the expression of $F(z)$ as in (3), $x$ and $y$ are the $\ell_1$ normalization of the first and second components of $z$ (see the definition of $g$ as in (2)), and the function $z \mapsto g(z)$ is badly behaved and may vary rapidly when $z \in \mathbb{R}_+^{d_1 + d_2}$ is close to the origin. We also write $\Pi_{\mathcal{Z}_\geq}(u)$ for the projection onto $\mathcal{Z}_\geq$ of the vector $u$: $\Pi_{\mathcal{Z}_\geq}(u) = \arg\min_{z' \in \mathcal{Z}_\geq} \|z' - u\|_2$. With these notations, Extragradient RM$^+$ (ExRM$^+$) and Smooth PRM$^+$ (SPRM$^+$) are defined in Algorithm 3 and in Algorithm 4.

---

**Algorithm 3** Extragradient RM$^+$ (ExRM$^+$)

1: **Input**: Step size $\eta \in (0, \frac{1}{L_F})$.
2: **Initialize**: $z^0 \in \mathcal{Z}$
3: **for** $t = 0, 1, \dots$ **do**
4: $\quad z^{t+1/2} = \Pi_{\mathcal{Z}_\geq}(z^t - \eta F(z^t))$
5: $\quad z^{t+1} = \Pi_{\mathcal{Z}_\geq}(z^t - \eta F(z^{t+1/2}))$

**Algorithm 4** Smooth PRM$^+$ (SPRM$^+$)

1: **Input**: Step size $\eta \in (0, \frac{1}{8L_F}]$.
2: **Initialize**: $z^{-1} = w^0 \in \mathcal{Z}$
3: **for** $t = 0, 1, \dots$ **do**
4: $\quad z^t = \Pi_{\mathcal{Z}_\geq}(w^t - \eta F(z^{t-1}))$
5: $\quad w^{t+1} = \Pi_{\mathcal{Z}_\geq}(w^t - \eta F(z^t))$

---

ExRM$^+$ is connected to the Extragradient (EG) algorithm (Korpelevich, 1976) and SPRM$^+$ is connected to the Optimistic Gradient (OG) algorithm (Popov, 1980; Rakhlin and Sridharan, 2013) (see Section 4 and Section 5 for details). Farina et al. (2023) show that ExRM$^+$ and SPRM$^+$ enjoy fast $O(\frac{1}{T})$ convergence (for the average of the iterates) for solving matrix games, but since the regret operator $F$ is *not* monotone, existing results for the convergence of EG and Optimistic Gradient on monotone games Cai et al. (2022) do not apply, and nothing is known about the last-iterate convergence properties of ExRM$^+$ and SPRM$^+$.

## 3 NON-CONVERGENCE OF RM$^+$, ALTERNATING RM$^+$, AND PRM$^+$

In this section, we show empirically that several existing variants of RM$^+$ may not converge in iterates. Specifically, we numerically investigate four algorithms—RM$^+$, alternating RM$^+$, PRM$^+$, and alternating PRM$^+$—on a simple $3 \times 3$ game matrix $A = [[3, 0, -3], [0, 3, -4], [0, 0, 1]]$ that has the unique Nash equilibrium $(x^\star, y^\star) = ([\frac{1}{12}, \frac{1}{12}, \frac{5}{6}], [\frac{1}{3}, \frac{5}{12}, \frac{1}{4}])$. The same instance was also used in (Farina et al., 2023) to illustrate the instability of PRM$^+$ and slow ergodic convergence of RM$^+$ and PRM$^+$. The results are shown in Figure 1. We observe that for RM$^+$, alternating RM$^+$, and PRM$^+$, the duality gap remains on the order of $10^{-1}$ even after $10^5$ iterations, which also suggests that these algorithms do not enjoy any *best-iterate* convergence properties. Our empirical findings are in line with Theorem 3 of Lee et al. (2021), who pointed out that RM$^+$ diverges on the rock-paper-scissors game. In contrast, alternating PRM$^+$ enjoys good last-iterate convergence properties on this instance. Overall, our empirical results suggest that RM$^+$, alternating RM$^+$, and PRM$^+$ all fail to converge in iterates, *even when the game has a unique Nash equilibrium*, a more regular and benign setting than the general case. To illustrate their non-converging properties, we provide the plots of the last iterates of RM$^+$, alternating RM$^+$, PRM$^+$ after $10^5$ iterations in Appendix C.

We complement our empirical non-convergence results by showing that RM$^+$ has asymptotic convergence under the restrictive assumption that the game has a *strict* NE. To our knowledge, this is the first positive last-iterate convergence result related to RM$^+$. In a strict NE $(x^\star, y^\star)$, $x^\star$ is the *unique* best-response to $y^\star$ and *vice versa*.

**Theorem 1** (Convergence of RM$^+$ to Strict NE). *If a matrix game has a* strict Nash equilibrium $(x^\star, y^\star)$, *RM$^+$ (Algorithm 1) converges in last-iterate, that is,* $\lim_{t \to \infty} \{(x^t, y^t)\} = (x^\star, y^\star)$.

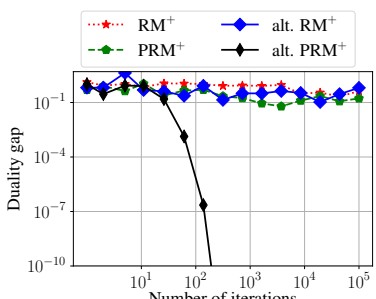

Figure 1: Duality gap of the current iterates generated by RM$^+$, PRM$^+$, and their alternating variants on the zero-sum game with payoff matrix $A = [[3, 0, -3], [0, 3, -4], [0, 0, 1]]$.

The assumption of strict NE in Theorem 1 cannot be weakened to the assumption of a unique, non-strict NE, as our empirical counterexample shows. Despite the isolated positive result given in Theorem 1 (under a very strong assumption that we do not expect to hold in practice), the negative empirical results from Figure 1 paint a bleak picture of the last-iterate convergence of unmodified regret-matching algorithms. This sets the stage for the rest of the paper, where we will show unconditional last-iterate convergence of SPRM$^+$ and ExRM$^+$.

## 4 CONVERGENCE PROPERTIES OF EXRM$^+$

In this section, we study ExRM$^+$, an algorithm proposed by (Farina et al., 2023). Although ExRM$^+$ lacks the no-regret property in the adversarial setting, understanding its performance is crucial due to its similarity to the important optimization algorithm EG. We show that ExRM$^+$ exhibits favorable last-iterate properties. We start by showing convergence in *duality gap*, and strengthen the result to *convergence in iterates* in Section 4.1. Then, in Section 4.2, we provide a concrete rate of $O(1/\sqrt{T})$ for the best iterate, based on which we finally show a linear last-iterate convergence rate using a restarting mechanism in Section 4.3. All omitted proofs for this section can be found in Appendix F. In Section 5, we study SPRM$^+$, a regret minimizer in adversarial settings, and show similar results.

ExRM$^+$ (Algorithm 3) is equivalent to the Extragradient (EG) algorithm of Korpelevich (1976) for solving a *variational inequality* VI($\mathcal{Z}_{\geq}, F$). For a closed convex set $\mathcal{S} \subseteq \mathbb{R}^n$ and an operator $G : \mathcal{S} \to \mathbb{R}^n$, the variational inequality problem VI($\mathcal{S}, G$) is to find $z \in \mathcal{S}$, such that $\langle G(z), z - z' \rangle \leq 0$ for all $z' \in \mathcal{S}$. We denote SOL($\mathcal{S}, G$) the solution set of VI($\mathcal{S}, G$). For any solution $z$ of VI($\mathcal{Z}_{\geq}, F$), the strategy profile after $\ell_1$-normalization $(x, y) = g(z)$ is a Nash equilibrium of the matrix game $A$. The last-iterate properties of EG are known in several settings, including:

1. If $G$ is Lipschitz and *pseudo monotone with respect to the solution set* SOL($\mathcal{S}, G$), i.e.,

$$\forall z^\star \in \text{SOL}(\mathcal{S}, G), \forall z \in \mathcal{S}, \langle G(z), z - z^\star \rangle \geq 0 \tag{4}$$

then iterates produced by EG converge to a solution of VI($\mathcal{S}, G$) (Facchinei and Pang, 2003, Ch. 12);

2. If $G$ is Lipschitz and *monotone*, i.e.

$$\forall \, z, z' \in \mathcal{S}, \langle G(z) - G(z'), z - z' \rangle \geq 0 \tag{5}$$

then iterates $\{z^t\}$ produced by EG have $O(\frac{1}{\sqrt{t}})$ last-iterate convergence such that $\langle G(z^t), z^t - z \rangle \leq O(\frac{1}{\sqrt{t}})$ for all $z \in \mathcal{S}$ (Golowich et al., 2020; Gorbunov et al., 2022a; Cai et al., 2022).

Unfortunately, these results do not apply directly to our case: although the operator $F$ (as defined in Equation (3)) is $L_F$-Lipschitz-continuous over $\mathcal{Z}_>$, it is not monotone or even pseudo monotone with respect to $\text{SOL}(\mathcal{Z}_\geq, F)$, as we show in Appendix D. However, $F$ satisfies the *Minty condition*:

$$\exists \, z^\star \in \text{SOL}(\mathcal{Z}_\geq, F), \forall \, z \in \mathcal{Z}_\geq, \langle F(z), z - z^\star \rangle \geq 0. \tag{6}$$

The Minty condition is weaker than pseudo monotonicity (4) (note the different quantifiers $\forall$ and $\exists$ for $z^\star$ in the two conditions). We now show that $F$ satisfies the Minty condition.

**Lemma 1.** *For any Nash equilibrium $z^\star \in \mathcal{Z}$, $\langle F(z), z - az^\star \rangle \geq 0$ holds for all $a \geq 1$.*

We are now ready to show convergence *in duality gap* for the iterates produced by ExRM⁺, which is weaker than *convergence in iterates*, studied in the next subsection. Our analysis follows from some results in Facchinei and Pang (2003), stated for the case of pseudomonotonicity, but extending for the Minty condition.

**Lemma 2** (Adapted from Lemma 12.1.10 in (Facchinei and Pang, 2003)). *Let $z^\star \in \mathcal{Z}_\geq$ be a point such that $\langle F(z), z - z^\star \rangle \geq 0$ for all $z \in \mathcal{Z}_\geq$. Let $\{z^t\}$ be the sequence produced by ExRM⁺. Then $\|z^{t+1} - z^\star\|^2 \leq \|z^t - z^\star\|^2 - (1 - \eta^2 L_F^2)\|z^{t+\frac{1}{2}} - z^t\|^2, \forall \, t \geq 0.$*

Lemma 2 shows that the sequence $\{z^t\}$ is bounded, so it has at least one limit point $\hat{z} \in \mathcal{Z}_\geq$ by compactness.[4] Using Lemma 2 and minor modifications to the proof of Theorem 12.1.11 in Facchinei and Pang (2003), the next lemma shows that every limit point $\hat{z}$ of $\{z^t\}$ lies in $\text{SOL}(\mathcal{Z}_\geq, F)$ and thus induces a Nash equilibrium. Moreover, $\lim_{t\to\infty} \|z^t - z^{t+1}\| = 0$ and $\lim_{t\to\infty} \text{DualityGap}(g(z^t)) = 0$.

**Lemma 3.** *Let $\{z^t\}$ be the iterates produced by ExRM⁺. Then the following holds:*

 *1. $\lim_{t\to\infty} \|z^t - z^{t+1}\| = 0$.*
 *2. If $\hat{z}$ is a limit point of $\{z^t\}$, then $\hat{z} \in \text{SOL}(\mathcal{Z}_\geq, F)$.*
 *3. $\lim_{t\to\infty} \text{DualityGap}(g(z^t)) = 0$.*

As shown in Lemma 3, for an operator that only satisfies the Minty condition, classical analysis yields asymptotic *convergence in duality gap*.

### 4.1 CONVERGENCE OF THE ITERATES

**Convergence in the Duality Gap Does Not Rule Out Cycling.** Convergence in the duality gap only implies that the trajectory $\{z^t\}$ converges to the set of solutions but **does not** guarantee *convergence in iterates*, i.e., that $\lim_{t\to\infty} z^t$ exists and is a solution. Even when combined with the conditions $\lim_{t\to\infty} \|z^t - z^{t+1}\| = 0$ and $\lim_{t\to\infty} \text{DualityGap}(g(z^t)) = 0$ (Lemma 3), we cannot conclude that the trajectory $\{z^t\}$ converges to a single point. It is possible for the trajectory to *cycle* around the set of Nash equilibria without ever *converging* to a single point. Specifically, these two conditions do not exclude the possibility that the per-iteration movement $\|z^t - z^{t+1}\|$ vanishes (e.g., at a rate $O(\frac{1}{t^a})$ for some $0 < a \leq 1$), while the total movement $\sum_{t=1}^\infty \|z^t - z^{t+1}\|$ diverges to infinity.

**Geometric Structure of Limit Points** To show the strong last-iterate convergence in the iterate property, *we make new observations on the structure of the solution set and provide characterizations of the learning dynamics' limit points.* First, the sequence $\{z^t\}$ has at least one limit point $\hat{z} \in \text{SOL}(\mathcal{Z}_\geq, F)$. If $\hat{z}$ is the unique limit point, then $\{z^t\}$ converges to $\hat{z}$. In the next proposition, we provide another condition on the limit point under which $\{z^t\}$ converges.

**Proposition 1.** *If the iterates $\{z^t\}$ produced by ExRM⁺ have a limit point $\hat{z}$ such that $\hat{z} = az^\star$ for $z^\star \in \mathcal{Z}$ and $a \geq 1$ (equivalently, colinear with a pair of strategies), then $\{z^t\}$ converges to $\hat{z}$.*

---

[4]A limit point of a sequence is the limit of one of its subsequences.

However, the condition of Proposition 1 may not hold in general, and we empirically observe in experiments that it is not uncommon for the limit point $\hat{z} = (R\hat{x}, Q\hat{y})$ to have $R \neq Q$. To proceed, we will use the observation that the only "bad" case that prevents us from proving convergence of $\{z^t\}$ is that $\{z^t\}$ has infinitely-many limit points (note that the number of solutions $|\mathrm{SOL}(\mathcal{Z}_>, F)|$ is indeed infinite). This is because if $\{z^t\}$ has a finite number of limit points, then since $\lim_{t\to\infty} \|z^{t+1} - z^t\| = 0$ (Lemma 3), it must have a unique limit point (see a formal proof in Proposition 3). In the following, to show that it is impossible that $\{z^t\}$ has infinitely many limit points, we first prove a lemma showing the structure of limit points of $\{z^t\}$. We illustrate this lemma in Figure 2.[5]

**Lemma 4** (Structure of Limit Points). *Let $\{z^t\}$ be the iterates produced by ExRM$^+$ and $z^\star \in \Delta^{d_1} \times \Delta^{d_2}$ be any Nash equilibrium of A. If $\hat{z}$ and $\tilde{z}$ are two limit points of $\{z^t\}$, then the following holds.*

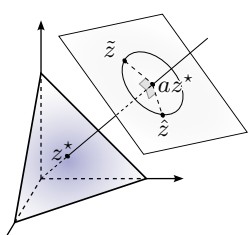

*1. $\|az^\star - \hat{z}\|^2 = \|az^\star - \tilde{z}\|^2$ for all $a \geq 1$.*
*2. $\|\hat{z}\|^2 = \|\tilde{z}\|^2$.*
*3. $\langle z^\star, \hat{z} - \tilde{z} \rangle = 0$.*

*Proof.* Let $\{k_i \in \mathbb{Z}\}$ be an increasing sequence of indices such that $\{z^{k_i}\}$ converges to $\hat{z}$ and $\{l_i \in \mathbb{Z}\}$ be an increasing sequence of indices such that $\{z^{l_i}\}$ converges to $\tilde{z}$. Let $\sigma : \{k_i\} \to \{l_i\}$ be a mapping that always maps $k_i$ to a larger index in $\{l_i\}$, i.e. $\sigma(k_i) > k_i$. Such a mapping clearly exists. Since Lemma 2 applies to $az^\star$ for any $a \geq 1$, we get

Figure 2: Pictorial illustration of Lemma 4.

$$\|az^\star - \hat{z}\|^2 = \lim_{i\to\infty} \|az^\star - z^{k_i}\|^2 \geq \lim_{i\to\infty} \|az^\star - z^{\sigma(k_i)}\|^2 = \|az^\star - \tilde{z}\|^2.$$

By symmetry, the other direction also holds. Thus $\|az^\star - \hat{z}\|^2 = \|az^\star - \tilde{z}\|^2$ for all $a \geq 1$. Expanding $\|az^\star - \hat{z}\|^2 = \|az^\star - \tilde{z}\|^2$ gives $\|\hat{z}\|^2 - \|\tilde{z}\|^2 = 2a\langle z^\star, \hat{z} - \tilde{z}\rangle$ for all $a \geq 1$. It implies that $\|\hat{z}\|^2 = \|\tilde{z}\|^2$ and $\langle z^\star, \hat{z} - \tilde{z}\rangle = 0$. $\square$

We are now ready to show that $\{z^t\}$ necessarily has a unique limit point.

**Lemma 5** (Unique limit point). *The sequence $\{z^t\}$ produced by ExRM$^+$ has a unique limit point.*

*Proof.* For the sake of contradiction, let $\hat{z} = (\widehat{R}\hat{x}, \widehat{Q}\hat{y})$ and $\tilde{z} = (\widetilde{R}\tilde{x}, \widetilde{Q}\tilde{y})$ be any two distinct limit points of $\{z^t\}$. We first prove a key equality $\widehat{R} + \widetilde{R} = \widehat{Q} + \widetilde{Q}$ by considering two cases.

• *Case 1: $\hat{z} = (\widehat{R}x^\star, \widehat{Q}y^\star)$ and $\tilde{z} = (\widetilde{R}x^\star, \widetilde{Q}y^\star)$ for a Nash equilibrium $z^\star = (x^\star, y^\star)$.* Part 2 of Lemma 4 gives $\|\hat{z}\|^2 = \|\tilde{z}\|^2 \Rightarrow (\widehat{R}^2 - \widetilde{R}^2)\|x^\star\|^2 = (\widetilde{Q}^2 - \widehat{Q}^2)\|y^\star\|^2$. Part 3 of Lemma 4 gives $\langle z^\star, \hat{z} - \tilde{z}\rangle = 0 \Rightarrow (\widehat{R} - \widetilde{R})\|x^\star\|^2 = (\widetilde{Q} - \widehat{Q})\|y^\star\|^2$. Combining the above two equalities with the fact that $\hat{z} \neq \tilde{z}$, we get $\widehat{R} + \widetilde{R} = \widehat{Q} + \widetilde{Q}$.

• *Case 2: $\hat{z} = (\widehat{R}\hat{x}, \widehat{Q}\hat{y})$ and $\tilde{z} = (\widetilde{R}\tilde{x}, \widetilde{Q}\tilde{y})$ for different Nash equilibria $(\hat{x}, \hat{y})$ and $(\tilde{x}, \tilde{y})$.* Since both $(\hat{x}, \hat{y})$ and $(\tilde{x}, \tilde{y})$ are Nash equilibrium (Lemma 3), we have $(\hat{x}, \tilde{y})$ is also a Nash equilibrium by exchangeability of Nash equilibria in zero-sum games. By Lemma 4, we have the following equalities

1. $\|\hat{z}\|^2 = \|\tilde{z}\|^2 \Rightarrow \widehat{R}^2\|\hat{x}\|^2 - \widetilde{R}^2\|\tilde{x}\|^2 = \widetilde{Q}^2\|\tilde{y}\|^2 - \widehat{Q}^2\|\hat{y}\|^2$
2. $0 = \langle(\hat{x}, \hat{y}), \hat{z} - \tilde{z}\rangle = \widehat{R}\|\hat{x}\|^2 - \widetilde{R}\langle\hat{x}, \tilde{x}\rangle + \widehat{Q}\|\hat{y}\|^2 - \widetilde{Q}\langle\hat{y}, \tilde{y}\rangle$
3. $0 = \langle(\tilde{x}, \tilde{y}), \hat{z} - \tilde{z}\rangle = \widehat{R}\langle\hat{x}, \tilde{x}\rangle - \widetilde{R}\|\tilde{x}\|^2 + \widehat{Q}\langle\hat{y}, \tilde{y}\rangle - \widetilde{Q}\|\tilde{y}\|^2$
4. $0 = \langle(\hat{x}, \tilde{y}), \hat{z} - \tilde{z}\rangle = \widehat{R}\|\hat{x}\|^2 - \widetilde{R}\langle\hat{x}, \tilde{x}\rangle + \widehat{Q}\langle\hat{y}, \tilde{y}\rangle - \widetilde{Q}\|\tilde{y}\|^2$

Combing the last three equalities gives

$$c := \widehat{R}\|\hat{x}\|^2 - \widetilde{R}\langle\hat{x}, \tilde{x}\rangle = \widehat{R}\langle\hat{x}, \tilde{x}\rangle - \widetilde{R}\|\tilde{x}\|^2, \quad \text{and} \quad -c = \widehat{Q}\|\hat{y}\|^2 - \widetilde{Q}\langle\hat{y}, \tilde{y}\rangle = \widehat{Q}\langle\hat{y}, \tilde{y}\rangle - \widetilde{Q}\|\tilde{y}\|^2.$$

Further combining the first equality gives

$$(\widehat{R} + \widetilde{R})c + (\widehat{Q} + \widetilde{Q})(-c)$$
$$= \widehat{R}(\widehat{R}\|\hat{x}\|^2 - \widetilde{R}\langle\hat{x}, \tilde{x}\rangle) + \widetilde{R}(\widehat{R}\langle\hat{x}, \tilde{x}\rangle - \widetilde{R}\|\tilde{x}\|^2) + \widehat{Q}(\widehat{Q}\|\hat{y}\|^2 - \widetilde{Q}\langle\hat{y}, \tilde{y}\rangle) + \widetilde{Q}(\widehat{Q}\langle\hat{y}, \tilde{y}\rangle - \widetilde{Q}\|\tilde{y}\|^2)$$

---

[5]We draw $z^\star$ in a simplex only as a simplified illustration—technically $z^\star$ should be from the Cartesian product of two simplices instead.

$$= \widehat{R}^2 \|\hat{x}\|^2 - \widetilde{R}^2 \|\tilde{x}\|^2 - \widetilde{Q}^2 \|\tilde{y}\|^2 + \widehat{Q}^2 \|\hat{y}\|^2 = 0.$$

If $c \neq 0$, then we get $\widehat{R} + \widetilde{R} = \widehat{Q} + \widetilde{Q}$. If $c = 0$, then $\widehat{R}\widetilde{R}\|\hat{x}\|^2\|\tilde{x}\|^2 = \widehat{R}\widetilde{R}\langle\hat{x}, \tilde{x}\rangle^2$ and $\widehat{Q}\widetilde{Q}\|\hat{y}\|^2\|\tilde{y}\|^2 = \widehat{Q}\widetilde{Q}\langle\hat{y}, \tilde{y}\rangle^2$ gives the equality condition of Cauchy-Schwarz inequality and we get $\hat{x} = \tilde{x}$ and $\hat{y} = \tilde{y}$. This reduced to Case 1 where we know $\widehat{R} + \widetilde{R} = \widehat{Q} + \widetilde{Q}$ holds. This completes the proof of $\widehat{R} + \widetilde{R} = \widehat{Q} + \widetilde{Q}$.

Note that $\widehat{R} + \widetilde{R} = \widehat{Q} + \widetilde{Q}$ must hold for any two distinct limit points $\hat{z}$ and $\tilde{z}$ of $\{z^t\}$. If $\widehat{R} = \widehat{Q}$, then $\hat{z} = \widehat{R}(\hat{x}, \hat{y})$ is colinear with a Nash equilibrium $(\hat{x}, \hat{y})$ and by Proposition 1, $\{z^t\}$ converges to $\hat{z}$ and $\hat{z}$ is the unique limit point. If $\widehat{R} \neq \widehat{Q}$, we argue that $\hat{z}$ and $\tilde{z}$ are the only two possible limit points. To see why, let us suppose that there exists another limit point $z = (Rx, Qy)$ distinct from $\hat{z}$ and $\tilde{z}$. If $R = \widehat{R}$, then by $R + \widetilde{R} = Q + \widetilde{Q}$, we know $Q = \widehat{Q}$. However, $R + \widehat{R} = 2\widehat{R} \neq 2\widehat{Q} = Q + \widehat{Q}$ gives a contradiction to the equality $\widehat{R} + \widetilde{R} = \widehat{Q} + \widetilde{Q}$ we just proved. If $R \neq \widehat{R}$, then combing

$$\widehat{R} + R = \widehat{Q} + Q, \quad \widetilde{R} + R = \widetilde{Q} + Q, \quad \widehat{R} + \widetilde{R} = \widehat{Q} + \widetilde{Q}$$

gives $\widehat{R} - \widehat{Q} = Q - R = \widetilde{R} - \widetilde{Q} = \widehat{Q} - \widehat{R}$. Thus, we have $\widehat{R} = \widehat{Q}$, and it contradicts the assumption that $\widehat{R} \neq \widehat{Q}$. Now we conclude that $\{z^t\}$ has at most two distinct limit points which means $\{z^t\}$ has a unique limit point given the fact that $\{z^t\}$ is bounded and $\lim_{t\to\infty} \|z^{t+1} - z^t\| = 0$. $\qquad\square$

Our main result now follows from Lemma 3 and Lemma 5. To our knowledge, this is the first last-iterate convergence result for a variational inequality problem satisfying *only* the Minty condition.

**Theorem 2** (Last-Iterate Convergence of ExRM$^+$). *Let $\{z^t\}$ be the sequence produced by ExRM$^+$, then $\{z^t\}$ converges to $z^* \in \mathcal{Z}_\geq$ with $g(z^*) = (x^*, y^*) \in \Delta^{d_1} \times \Delta^{d_2}$ being a Nash equilibrium.*

## 4.2 Best-Iterate Convergence Rate of ExRM$^+$

We now prove an $O(\frac{1}{\sqrt{T}})$ best-iterate convergence rate of ExRM$^+$. The following key lemma relates the duality gap of a pair of strategies $(x^{t+1}, y^{t+1})$ and the distance $\|z^{t+\frac{1}{2}} - z^t\|$.

**Lemma 6.** *Let $\{z^t\}$ be iterates produced by ExRM$^+$ and $(x^{t+1}, y^{t+1}) = g(z^{t+1})$. Then* $\text{DualityGap}(x^{t+1}, y^{t+1}) \leq \frac{12\|z^{t+\frac{1}{2}} - z^t\|}{\eta}$.

Now combining Lemma 2 and Lemma 6, we conclude the following best-iterate convergence rate.

**Theorem 3.** *Let $\{z^t\}$ be the sequence produced by ExRM$^+$ with initial point $z^0$. Then for any Nash equilibrium $z^*$, any $T \geq 1$, there exists $t \leq T$ with $(x^t, y^t) = g(z^t)$ and, for $\eta = \frac{1}{\sqrt{2}L_F}$,* $\text{DualityGap}(x^t, y^t) \leq \frac{24L_F\|z^0 - z^*\|}{\sqrt{T}}$.

*Proof.* Fix a NE $z^*$. From Lemma 2, $\sum_{t=0}^{T-1} \|z^{t+\frac{1}{2}} - z^t\|^2 \leq \frac{\|z^0 - z^*\|^2}{1 - \eta^2 L_F^2}$. This implies that $\exists\, t \leq T-1$ such that $\|z^{t+\frac{1}{2}} - z^t\| \leq \frac{\|z^0 - z^*\|}{\sqrt{T}\sqrt{1 - \eta^2 L_F^2}}$. We then get the result by applying Lemma 6. $\qquad\square$

## 4.3 Linear Last-Iterate Convergence for ExRM$^+$ with Restarts

In this section, based on the best-iterate convergence result from the last section, we further provide a simple restarting mechanism under which ExRM$^+$ enjoys linear last-iterate convergence. To show this, we recall that zero-sum matrix games satisfy the following *metric subregularity* condition.

**Proposition 2** (Metric Subregularity (Wei et al., 2021)). *Let $A \in \mathbb{R}^{d_1 \times d_2}$ be a matrix game. There exists a constant $c > 0$ (only depending on $A$) such that for any $z = (x, y) \in \Delta^{d_1} \times \Delta^{d_2}$, $\text{DualityGap}(x, y) \geq c\|z - \Pi_{\mathcal{Z}^*}[z]\|$ where $\mathcal{Z}^*$ denotes the set of all Nash equilibria.*

Together with Theorem 3 (with $\eta = \frac{1}{\sqrt{2}L_F}$), we obtain that for any $T \geq 1$, there exists $1 \leq t \leq T$ such that $\|(x^t, y^t) - \Pi_{\mathcal{Z}^*}[(x^t, y^t)]\| \leq \frac{\text{DualityGap}(x^t, y^t)}{c} \leq \frac{24L_F}{c\sqrt{T}} \cdot \|z^0 - \Pi_{\mathcal{Z}^*}[z^0]\|$. This inequality further implies that if $T \geq \frac{48^2 L_F^2}{c^2}$, then there exists $1 \leq t \leq T$ such that

$$\left\|(x^t, y^t) - \Pi_{\mathcal{Z}^*}[(x^t, y^t)]\right\| \leq \frac{1}{2}\|z^0 - \Pi_{\mathcal{Z}^*}[z^0]\|.$$

Therefore, after at most a constant number of iterations (smaller than $\frac{48^2 L_F^2}{c^2}$), the distance of the best-iterate $(x^t, y^t)$ to the equilibrium set $\mathcal{Z}^\star$ is halved compared to that of the initial point. If we could somehow identify this best iterate, then we just need to restart the algorithm with this best iterate as the next initial strategy. Repeating this would then lead to a linear last-iterate convergence. The issue in this argument above is that we cannot exactly identify the best iterate since $\|(x^t, y^t) - \Pi_{\mathcal{Z}^\star}[(x^t, y^t)]\|$ is unknown. However, we use $\|z^{t+\frac{1}{2}} - z^t\|$ as a proxy, since

$$\left\| (x^t, y^t) - \Pi_{\mathcal{Z}^\star}[(x^t, y^t)] \right\| \leq \frac{1}{c} \operatorname{DualityGap}(x^t, y^t) \leq \frac{12\|z^{t+\frac{1}{2}} - z^t\|}{c\eta}$$

by Lemma 6. This motivates the design of Restarting ExRM+ (RS-ExRM$^+$, Algorithm 5), which restarts for the $k$-th time if $\|z^{t+\frac{1}{2}} - z^t\|$ is less than $O(\frac{1}{2^k})$. Importantly, RS-ExRM$^+$ does not require knowing the value of $c$, the constant in the metric subregularity condition, which can be hard to compute. The main result of this section is the following linear convergence rates of RS-ExRM$^+$.

| **Algorithm 5** Restarting ExRM$^+$ (RS-ExRM$^+$) | **Algorithm 6** Restarting SPRM$^+$ (RS-SPRM$^+$) |
|---|---|
| 1: **Input**: Step size $\eta \in (0, \frac{1}{L_F})$, $\rho > 0$. | 1: **Input**: Step size $\eta \in (0, \frac{1}{8L_F}]$. |
| 2: **Initialize**: $z^0 \in \mathcal{Z}$, $k = 1$ | 2: **Initialize**: $z^{-1} = w^0 \in \mathcal{Z}$, $k = 1$ |
| 3: **for** $t = 0, 1, \dots$ **do** | 3: **for** $t = 0, 1, \dots$ **do** |
| 4: $\quad z^{t+1/2} = \Pi_{\mathcal{Z}_\geq}\left(z^t - \eta F(z^t)\right)$ | 4: $\quad z^t = \Pi_{\mathcal{Z}_\geq}\left(w^t - \eta F(z^{t-1})\right)$ |
| 5: $\quad z^{t+1} = \Pi_{\mathcal{Z}_\geq}\left(z^t - \eta F(z^{t+1/2})\right)$ | 5: $\quad w^{t+1} = \Pi_{\mathcal{Z}_\geq}\left(w^t - \eta F(z^t)\right)$ |
| 6: $\quad$ **if** $\|z^{t+1/2} - z^t\| \leq \rho/2^k$ **then** | 6: $\quad$ **if** $\|w^{t+1} - z^t\| + \|w^t - z^t\| \leq 8/2^k$ **then** |
| 7: $\quad\quad z^{t+1} \leftarrow g(z^{t+1}) \in \Delta^{d_1} \times \Delta^{d_2}$ | 7: $\quad\quad z^t, w^{t+1} \leftarrow g(w^{t+1}) \in \Delta^{d_1} \times \Delta^{d_2}$ |
| 8: $\quad\quad k \leftarrow k + 1$ | 8: $\quad\quad k \leftarrow k + 1$ |

**Theorem 4** (Linear Last-Iterate Convergence of RS-ExRM$^+$). *Let $\{z^t\}$ be the sequence produced by RS-ExRM$^+$ with constant step size $\eta$ and let $\rho = \frac{4}{\sqrt{1-\eta^2 L_F^2}}$. For any $t \geq 1$, the iterate $(x^t, y^t) = g(z^t)$ satisfies $\operatorname{DualityGap}(x^t, y^t) \leq c \cdot \alpha \cdot (1 - \beta)^t$ where $\alpha = \frac{576}{c^2 \eta^2 (1 - \eta^2 L_F^2)}$ and $\beta = \frac{1}{3(1+\alpha)}$.*

*Proof sketch.* Let $t_k$ be the iteration where the $k$-th restart happens. From the restart condition and Lemma 6, at iteration $t_k$, the duality gap of $(x^{t_k}, y^{t_k})$ and its distance to $\mathcal{Z}^\star$ is at most $O(\frac{1}{2^k})$. For iterate $t \in [t_k, t_{k+1}]$ at which the algorithm does not restart, we can use Theorem 3 to show that its performance is not much worse than that of $(x^{t_k}, y^{t_k})$. Then we prove $t_{k+1} - t_k$ is upper bounded by a constant for every $k$, which leads to a linear last-iterate convergence rate for all iterations $t \geq 1$. $\quad\square$

## 5 LAST-ITERATE CONVERGENCE OF SPRM$^+$

In this section we study SPRM$^+$ (Algorithm 4), an important variant of RM$^+$ based on OG. For the sake of conciseness, we only state the main results here. The proofs are presented in Appendix G.

**Theorem 5** (Asymptotic Last-Iterate Convergence of SPRM$^+$). *Let $\{w^t\}$ and $\{z^t\}$ be the sequences produced by SPRM$^+$, then $\{w^t\}$ and $\{z^t\}$ are bounded and both converge to $z^\star \in \mathcal{Z}_\geq$ with $g(z^\star) = (x^\star, y^\star) \in \Delta^{d_1} \times \Delta^{d_2}$ being a Nash equilibrium of the matrix game $A$.*

**Theorem 6** (Best-Iterate Convergence Rate of SPRM$^+$). *Let $\{w^t\}$ and $\{z^t\}$ be the sequences produced by SPRM$^+$. For any Nash equilibrium $z^\star$, any $T \geq 1$, there exists $1 \leq t, t' \leq T$ such that the iterate $g(w^t), g(z^{t'}) \in \Delta^{d_1} \times \Delta^{d_2}$ satisfy $\operatorname{DualityGap}(g(w^t)) \leq \frac{10\|w^0 - z^\star\|}{\eta} \frac{1}{\sqrt{T}}$, and $\operatorname{DualityGap}(g(z^t)) \leq \frac{18\|w^0 - z^\star\|}{\eta} \frac{1}{\sqrt{T}}$.*

We apply the idea of restarting to SPRM$^+$ to design a new algorithm called Restarting SPRM$^+$ (RS-SPRM$^+$; see Algorithm 6) with provable linear last-iterate convergence.

**Theorem 7** (Linear Last-Iterate Convergence of RS-SPRM$^+$). *Let $\{w^t\}$ and $\{z^t\}$ be the sequences produced by RS-SPRM$^+$ with constant step size $\eta$. Let $\alpha = \frac{400}{c^2 \eta^2}$ and $\beta = \frac{1}{3(1+\alpha)}$. For $t \geq 1$, the iterates $g(w^t), g(z^t) \in \mathcal{Z}$ satisfy $\operatorname{DualityGap}(g(w^t)) \leq c \cdot \alpha \cdot (1 - \beta)^t$ and $\operatorname{DualityGap}(g(z^t)) \leq 2c \cdot \alpha \cdot (1 - \beta)^t$.*

Figure 3: Empirical performances of several algorithms on the $3 \times 3$ matrix game (left plot), Kuhn poker and Goofspiel (center plots), and random instances (right plot).

## 6    NUMERICAL EXPERIMENTS

The goal in this section is to verify numerically our theoretical findings from the previous sections, pertaining to the lack of convergence properties of RM$^+$, PRM$^+$ and to the convergence properties of ExRM$^+$ and SPRM$^+$. For the sake of completeness, we also show the performances of the extragradient algorithm (EG) and optimistic gradient descent (OG) Cai et al. (2022). We use the $3 \times 3$ matrix game instance from Section 3, the normal-form representations of Kuhn poker and Goofspiel, as well as 25 random matrix games of size $(d_1, d_2) = (10, 15)$ (for which we average the duality gaps across the instances and show the associated confidence intervals). More details on the games and on EG and OG can be found in Appendix H. For the algorithms requiring a stepsize (ExRM$^+$, SPRM$^+$, EG and OG), we choose the best stepsizes $\eta \in \{1, 10^{-1}, 10^{-2}, 10^{-3}, 10^{-4}\}$. We initialize all algorithms at $((1/d_1)\mathbf{1}_{d_1}, (1/d_2)\mathbf{1}_{d_2})$. In every iteration, we plot the duality gap of $(x^t, y^t)$ for RM$^+$, PRM$^+$, and alternating PRM$^+$; the duality gap of $g(z^t)$ for ExRM$^+$; the duality gap of $g(w^t)$ for SPRM$^+$; and the duality gap of $z^t$ in EG and OG. The results are shown in Figure 3. To avoid issues with numerical precisions, we set a lower threshold of $10^{-10}$ for the duality gaps displayed in all the figures in this paper, including Figure 1 and the additional numerical results in the appendices. For the small matrix game, alternating PRM$^+$, ExRM$^+$, and SPRM$^+$ achieve machine precision after $10^3$ iterations (while other RM-based algorithms stay around $10^{-1}$ as discussed earlier), slightly faster than EG and OG. On Kuhn poker, PRM$^+$ and alternating PRM$^+$ have faster convergence early, but later all algorithms perform on par (except RM$^+$). On Goofspiel, all algorithms (except RM$^+$) have comparable performance after $10^5$ iterations, except EG which achieves machine precision. Finally, on random instances, the last iterate performance of ExRM$^+$, SPRM$^+$, and EG vastly outperform RM$^+$, PRM$^+$, and alternating RM$^+$, but we note that alternating PRM$^+$ outperforms all other algorithms. OG appears to be slow on these instances. We include additional experiments studying the best-iterate performances in Appendix H.4. They corroborate our theoretical findings of a $O(1/\sqrt{T})$ best-iterate convergence rates for ExRM$^+$ (Theorem 3) and SPRM$^+$ (Theorem 6).

Overall, these empirical results are consistent with, and validate, our theoretical findings of ExRM$^+$ and SPRM$^+$. That said, understanding the good practical performance of alternating PRM$^+$ (an algorithm for which neither ergodic nor last-iterate convergence guarantees are known) remains open. We also numerically investigate the impact of restarting for RS-ExRM$^+$ and RS-SPRM$^+$ (see Appendix H), where we note that both algorithms achieve linear last-iterate convergence (Theorem 4 and 7) and are faster than their non-restarting counterparts on some instances.

## 7    CONCLUSIONS

In this paper, we investigate the last-iterate convergence properties of regret-matching algorithms, a class of popular methods for equilibrium computation in games. Despite these methods enjoying strong *average* performance in practice, we show that, unfortunately, many practically-used variants might not converge. Motivated by these findings, we set out to investigate variants with provable last-iterate convergence, establishing a suite of new results by using techniques from the literature on variational inequalities. For a restarted variant of these algorithms, we were able to prove, for the first time for regret matching algorithms, linear last-iterate convergence rates. Several questions remain open, including giving concrete rates for (non-restarted) ExRM$^+$, SPRM$^+$, and alternating PRM$^+$. Another open problem is understanding if our results extend to the case of solving EFGs with RM$^+$ via the CFR framework. However, analyzing RM$^+$ behavior in the CFR setup is much more difficult because even in the two-player zero-sum case, the variational inequality characterization must be applied separately at every decision point.

ACKNOWLEDGEMENTS

Yang Cai was supported by the NSF Awards CCF-1942583 (CAREER) and CCF-2342642. Gabriele Farina was supported by the NSF Award CCF-2443068 (CAREER). Julien Grand-Clément was supported by Hi! Paris and Agence Nationale de la Recherche (Grant 11-LABX-0047). Christian Kroer was supported by the Office of Naval Research awards N00014-22-1-2530 and N00014-23-1-2374, and the National Science Foundation awards IIS-2147361 and IIS-2238960. Haipeng Luo was supported by the National Science Foundation award IIS-1943607. Weiqiang Zheng was supported by the NSF Awards CCF-1942583 (CAREER), CCF-2342642, and a Research Fellowship from the Center for Algorithms, Data, and Market Design at Yale (CADMY).

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

## CONTENTS

## A   ALTERNATING VARIANTS OF RM$^+$

We describe the alternating variants of RM$^+$ and Predictive RM$^+$ in Algorithm 7 and Algorithm 8.

| **Algorithm 7** Alternating RM$^+$ (alt. RM$^+$) | **Algorithm 8** Alternating Predictive RM$^+$ (alt. PRM$^+$) |
|---|---|
| 1: **Initialize**: $(R_x^0, R_y^0) = \mathbf{0}$, $(x^0, y^0) \in \mathcal{Z}$ | 1: **Initialize**: $(R_x^0, R_y^0) = \mathbf{0}$, $(x_0, y_0) \in \mathcal{Z}$ |
| 2: **for** $t = 0, 1, \dots$ **do** | 2: **for** $t = 0, 1, \dots$ **do** |
| 3:     $R_x^{t+1} = [R_x^t - f(x^t, Ay^t)]^+$ | 3:     $R_x^{t+1} = [R_x^t - f(x^t, Ay^t)]^+$ |
| 4:     $x^{t+1} = \frac{R_x^{t+1}}{\|R_x^{t+1}\|_1}$ | 4:     $x^{t+1} = \frac{[R_x^{t+1} - f(x^t, Ay^t)]^+}{\|[R_x^{t+1} - f(x^t, Ay^t)]^+\|_1}$ |
| 5:     $R_y^{t+1} = [R_y^t + f(y^t, A^\top x^{t+1})]^+$ | 5:     $R_y^{t+1} = [R_y^t + f(y^t, A^\top x^{t+1})]^+$ |
| 6:     $y^{t+1} = \frac{R_y^{t+1}}{\|R_y^{t+1}\|_1}$ | 6:     $y^{t+1} = \frac{[R_y^{t+1} + f(y^t, A^\top x^{t+1})]^+}{\|[R_y^{t+1} + f(y^t, A^\top x^{t+1})]^+\|_1}$ |

## B   NON-LIPSCHTIZNESS OF THE REGRET OPERATOR $F$

Here, we show that the operator $F$ defined in (3) is not $L$-Lipschitz continuous over $\mathbb{R}_+^{d_1+d_2}$ for any $L > 0$. Take any point $z, z' \in \Delta^{d_1} \times \Delta^{d_2}$ and let $c := \|F(z) - F(z')\|$. Note that by definition of $F$ we have $F(a \cdot z) = F(z)$ for any $a > 0$. Therefore, $\|F(az) - F(az')\| = \|F(z) - F(z')\| = c$ for any $a > 0$, yet we can choose $a > 0$ small enough so that $\|az - az'\| = a\|z - z'\| < c/L$, which shows that $F$ cannot be $L$-Lipschitz continuous, otherwise we would obtain $\|F(az) - F(az')\| \le L\|az - az'\| < c$ which is a contradiction.

## C   DIVERGING TRAJECTORIES OF RM$^+$, ALTERNATING RM$^+$ AND PREDICTIVE RM$^+$

In this section we plot the last 2000 iterations (out of $10^5$ iterations) of RM$^+$, alternating RM$^+$ and Predictive RM$^+$ for solving the matrix game from Figure 1. The results are presented in Figure 4,

Figure 5 and Figure 6. Since $x_t \in \Delta^3$ we only plot the first two components of $x^t$ and similarly for $y^t \in \Delta^3$.

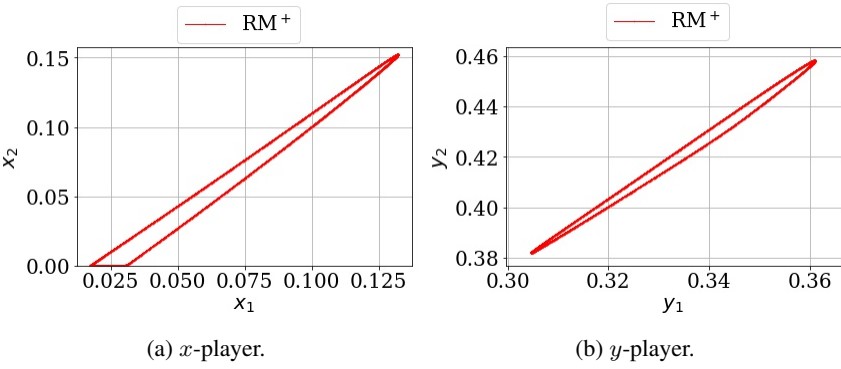

(a) $x$-player.  (b) $y$-player.

Figure 4: Last 2000 iterates of Regret Matching$^+$ after $10^5$ iterations for solving the matrix game from Figure 1.

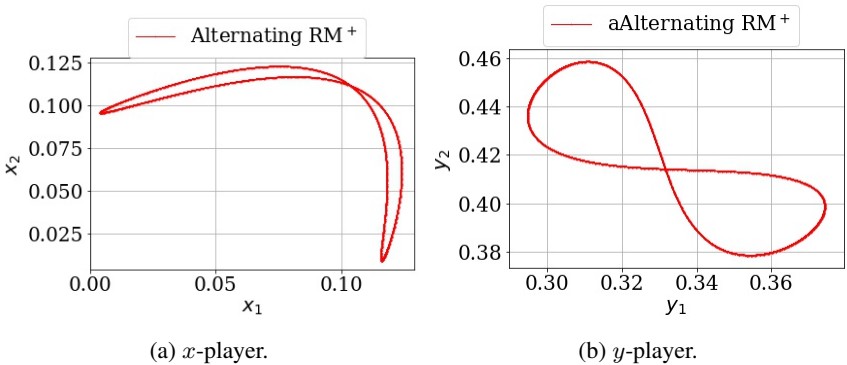

(a) $x$-player.  (b) $y$-player.

Figure 5: Last 2000 iterates of Alternating Regret Matching$^+$ after $10^5$ iterations for solving the matrix game from Figure 1.

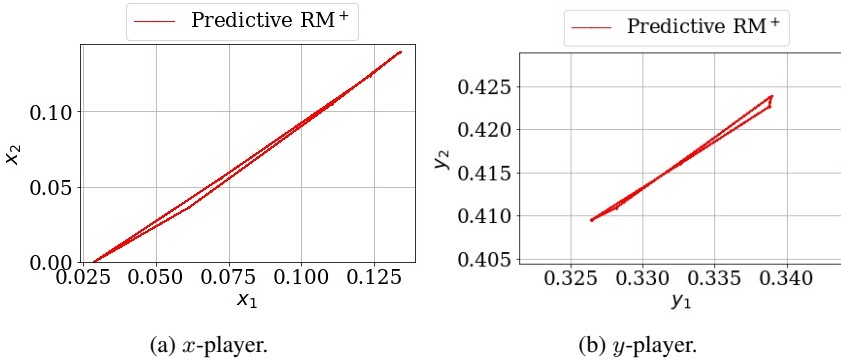

(a) $x$-player.  (b) $y$-player.

Figure 6: Last 2000 iterates of Predictive RM$^+$ after $10^5$ iterations for solving the matrix game from Figure 1.

# D USEFUL PROPOSITIONS

**Lemma 7** (Solution to VI Corresponds to Nash Equilibrium). *If $z \in SOL(\mathcal{Z}_\geq, F)$, then $(x, y) = g(z)$ is a Nash equilibrium of the matrix game $A$.*

*Proof.* Since $z \in SOL(\mathcal{Z}_\geq, F)$ is a solution, we have $z = \Pi_{\mathcal{Z}_\geq}[z - \eta F(z)]$ for any $\eta > 0$. Then by Lemma 11, DualityGap($g(z)$) $= 0$. This implies $(x, y) = g(z)$ is a Nash equilibrium. □

In the following two lemmas, we show that the *gradient operator* of a matrix game is always monotone, whereas its *regret operator* (used in RM-type algorithm) is not monotone, or even pseudo monotone with respect to the solution set.

**Lemma 8** (Gradient Operator is Monotone). *For any matrix game $A \in \mathbb{R}^{d_1 \times d_2}$, its gradient operator $G : \mathcal{Z} = \Delta^{d_1} \times \Delta^{d_2} \to \mathbb{R}^{d_1} \times \mathbb{R}^{d_2}$, defined as $G(z) = [Ay, -A^\top x]^\top$ for each $z = (x, y) \in \mathcal{Z}$, is monotone.*

*Proof.* For any two point $z = (x, y)$ and $z' = (x', y')$, we have

$$\langle G(z) - G(z'), z - z' \rangle = (x - x')^\top Ay - x^\top A(y - y') - (x - x')^\top Ay' + x'^\top A(y - y') = 0.$$

Thus $G$ is monotone over $\mathcal{Z}$. □

**Lemma 9** (Regret Operator can be Non-Monotone). *There exists a matrix game $A \in \mathbb{R}^{d_1 \times d_2}$ whose regret operator $F : \mathcal{Z}_\geq = \Delta^{d_1}_\geq \times \Delta^{d_2}_\geq \to \mathbb{R}$, defined as for each $z = (Rx, Qy)$ with $x, y \in \mathcal{Z} = \Delta^{d_1} \times \Delta^{d_2}$*

$$F(z) = F((Rx, Qy)) = \begin{bmatrix} f(x, Ay) \\ f(y, -A^\top x) \end{bmatrix} = \begin{bmatrix} Ay - x^\top Ay \cdot \mathbf{1}_{d_1} \\ -A^\top x + x^\top Ay \cdot \mathbf{1}_{d_2} \end{bmatrix},$$

*is not monotone, or even pseudo monotone w.r.t the solution set $SOL(\mathcal{Z}_\geq, F)$.*

*Proof.* We consider a matrix $A = [[3, 1], [4, 5]]$ whose unique Nash equilibrium is $(x^*, y^*) = ([1, 0]^\top, [1, 0]^\top)$. We define

- $z = (x, y) = ([0, 1]^\top, [0, 1]^\top)$;
- $z' = (x^*, 5y^*)$.

Then we have
$$\langle F(z) - F(z'), z - z' \rangle = -4 < 0.$$

Thus $F$ is not monotone over $\mathcal{Z}_\geq$. Moreover, $z'$ belongs to $SOL(\mathcal{Z}_\geq, F)$ and

$$\langle F(z), z - z' \rangle = -1 < 0.$$

Thus $F$ is not pseudo monotone with respect to the solution set $SOL(\mathcal{Z}_\geq, F)$. This also proves the equivalent claim that not all solutions in $SOL(\mathcal{Z}_\geq, F)$ satisfies the Minty condition (6). □

The rest of this section consists in the proof of Proposition 3.

**Proposition 3.** *Let $\mathcal{S} \subseteq \mathbb{R}^n$ be a compact convex set and $\{z^t \in \mathcal{S}\}$ be a sequence of points in $\mathcal{S}$ such that $\lim_{t \to \infty} \|z^t - z^{t+1}\| = 0$, then one of the following is true.*

1. *The sequence $\{z^t\}$ has infinite limit points.*

2. *The sequence $\{z^t\}$ has a unique limit point.*

*Proof.* We denote a closed Euclidean ball in $\mathbb{R}^n$ as $B(x, \delta) := \{x' \in \mathbb{R}^n : \|x' - x\| \leq \delta\}$. For the sake of contradiction, assume that neither of the two conditions are satisfied. This implies that the sequence $\{z^t\}$ has a finite number ($\geq 2$) of limit points. Then there exists $\epsilon > 0$ and two distinct limit points $\hat{z}_1$ and $\hat{z}_2$ such that

1. $\|\hat{z}_1 - \hat{z}_2\| \geq \epsilon$;

2. $\hat{z}_1$ is the only limit point of $\{z^t\}$ within $B(\hat{z}_1, \epsilon/2) \cap \mathcal{S}$;

3. $\hat{z}_2$ is the only limit point of $\{z^t\}$ within $B(\hat{z}_2, \epsilon/2) \cap \mathcal{S}$.

Given that $\lim_{t\to\infty} \|z^t - z^{t+1}\| = 0$, there exists $T > 0$ such that for any $t > T$, $\|z^t - z^{t+1}\| \leq \frac{\epsilon}{4}$. Now since $\hat{z}_1$ and $\hat{z}_2$ are limit points of $\{z^t\}$, there exists sub-sequences $\{z^{t_k}\}$ and $\{z^{t_j}\}$ that converges to $\hat{z}_1$ and $\hat{z}_2$, respectively. This means that there are infinite points from $\{z^{t_k}\}$ within $B(\hat{z}_1, \epsilon/4) \cap \mathcal{S}$ and infinite points from $\{z^{t_j}\}$ within $B(\hat{z}_2, \epsilon/4) \cap \mathcal{S}$. However, for $t > T$, there must also be infinite points from $\{z^t\}$ within the region $(B(\hat{z}_1, \epsilon/2) - B(\hat{z}_1, \epsilon/4)) \cap \mathcal{S}$. This implies there exists another limit point within $B(\hat{z}_1, \epsilon/2) \cap \mathcal{S}$ and contradicts the assertion that $\hat{z}_1$ is the only limit point within $B(\hat{z}_1, \epsilon/2) \cap \mathcal{S}$. □

## E  PROOF OF THEOREM 1

We remark that Theorem 1's proof largely follows from (Meng et al., 2023) with the key observation that the properties of strict NE suffices to adapt the analysis in (Meng et al., 2023) to zero-sum games (not strongly-convex-strongly-concave games) with a strict NE.

*Proof of Theorem 1.* We will use an equivalent update rule of RM$^+$ (Algorithm 1) proposed by Farina et al. (2021): for all $t \geq 0$, $z^{t+1} = \Pi_{\mathbb{R}_{\geq 0}^{d_1+d_2}}[z^t - \eta F(z^t)]$ where we use $z^t$ to denote $(R_x^t, R_y^t)$ and $F$ as defined in (3). The step size $\eta > 0$ only scales the $z^t$ vector and produces the same strategies $\{(x^t, y^t) \in \Delta^{d_1} \times \Delta^{d_2}\}$ as Algorithm 1. In the proof, we also use the following property of the operator $F$, which is weaker than Lipschitzness.

**Claim 1** (Adapted from Lemma 5.7 in (Farina et al., 2023))**.** *For a matrix game $A \in \mathbb{R}^{d_1 \times d_2}$, let $L_1 = \|A\|_{op}\sqrt{6\max\{d_1, d_2\}}$ with $\|A\|_{op} := \sup\{\|Av\|_2/\|v\|_2 : v \in \mathbb{R}^{d_2}, v \neq \mathbf{0}\}$. Then for any $z, z' \in \mathbb{R}_{\geq 0}^{d_1+d_2}$, it holds that $\|F(z) - F(z')\| \leq L_1\|g(z) - g(z')\|$.*

For any $t \geq 1$, since $z^{t+1} = \Pi_{\mathbb{R}_{\geq 0}^{d_1+d_2}}[z^t - \eta F(z^t)]$, we have for any $z \in \mathbb{R}_{\geq 0}^{d_1+d_2}$

$$\langle z^t - \eta F(z^t) - z^{t+1}, z - z^{t+1} \rangle \leq 0.$$
$$\Rightarrow \langle z^t - z^{t+1}, z - z^t \rangle \leq -\langle z^t - z^{t+1}, z^t - z^{t+1} \rangle + \langle \eta F(z^t), z - z^{t+1} \rangle. \tag{7}$$

By three-point identity, we have

$$\|z^{t+1} - z\|^2 = \|z^t - z\|^2 + \|z^{t+1} - z^t\|^2 + 2\langle z^t - z^{t+1}, z - z^t \rangle$$
$$\leq \|z^t - z\|^2 - \|z^{t+1} - z^t\|^2 + 2\langle \eta F(z^t), z - z^{t+1} \rangle. \tag{by (7)}$$

We define $k^t := \arg\min_{k\geq 1} \|z^t - kz^\star\|$. With $z = k^t z^\star$, the above inequality is equivalent to

$$\|z^{t+1} - k^t z^\star\|^2 + 2\eta\langle F(z^\star), z^{t+1} \rangle$$
$$\leq \|z^t - k^t z^\star\|^2 + 2\eta\langle F(z^\star), z^t \rangle - \|z^{t+1} - z^t\|^2 + 2\eta\langle F(z^t) - F(z^\star), z^t - z^{t+1} \rangle$$
$$+ 2\eta\langle F(z^t), k^t z^\star \rangle \qquad\qquad \text{(We use } \langle F(z^t), z^t \rangle = 0)$$
$$\leq \|z^t - k^t z^\star\|^2 + 2\eta\langle F(z^\star), z^t \rangle + \eta^2\|F(z^t) - F(z^\star)\|^2 + 2\eta\langle F(z^t), k^t z^\star \rangle$$
$$\qquad\qquad\qquad\qquad (-a^2 + 2ab \leq b^2)$$
$$\leq \|z^t - k^t z^\star\|^2 + 2\eta\langle F(z^\star), z^t \rangle + \eta^2 L_1^2\|(x^t, y^t) - z^\star\|^2 + 2\eta\langle F(z^t), k^t z^\star \rangle.$$

Since $z^\star = (x^\star, y^\star)$ is a strict Nash equilibrium, by Lemma 10, there exists a constant $c > 0$ such that

$$2\eta\langle F(z^t), k^t z^\star \rangle = -2\eta k^t\big((x^t)^\top A y^\star - (x^\star)^\top A y^t\big)$$
$$\leq -2\eta c k^t\|(x^t, y^t) - z^\star\|^2$$
$$\leq -2\eta c\|(x^t, y^t) - z^\star\|^2,$$

where in the last inequality, we use $k^t \geq 1$ by definition. Thus, combining the above two inequalities gives

$$\left\|z^{t+1} - k^t z^\star\right\|^2 + 2\eta\langle F(z^\star), z^{t+1}\rangle \leq \left\|z^t - k^t z^\star\right\|^2 + 2\eta\langle F(z^\star), z^t\rangle + (\eta^2 L_1^2 - 2\eta c)\left\|(x^t, y^t) - z^\star\right\|^2.$$

Using definition of $k^{t+1}$ and choosing $\eta < \frac{c}{L_1^2}$, we have $2\eta c - \eta^2 L_1^2 > 0$ and

$$(2\eta c - \eta^2 L_1^2)\left\|(x^t, y^t) - z^\star\right\|^2 \leq \left\|z^t - k^t z^\star\right\|^2 + 2\eta\langle F(z^\star), z^t\rangle - \left\|z^{t+1} - k^{t+1}z^\star\right\|^2 - 2\eta\langle F(z^\star), z^{t+1}\rangle.$$

Telescoping the above inequality for $t \in [1, T]$ gives

$$(2\eta c - \eta^2 L_1^2) \sum_{t=1}^{T} \left\|(x^t, y^t) - z^\star\right\|^2 \leq \left\|z^1 - k^1 z^\star\right\|^2 + 2\eta\langle F(z^\star), z^1\rangle < +\infty$$

It implies that the sequence $\{(x^t, y^t)\}$ converges to $z^\star$. Moreover, it also gives a $O(\frac{1}{\sqrt{T}})$ best-iterate convergence rate with respect to $\|(x^t, y^t) - z^\star\|^2$. $\qquad\square$

**Lemma 10.** *If a matrix game $A$ has a strict Nash equilibrium $(x^\star, y^\star)$, then there exists a constant $c > 0$ such that for any $(x, y) \in \Delta^{d_1} \times \Delta^{d_2}$,*

$$x^\top A y^\star - (x^\star)^\top A y \geq c\|(x, y) - (x^\star, y^\star)\|^2.$$

*Proof.* Since $(x^\star, y^\star)$ is strict, it is also the unique and pure Nash equilibrium. we denote the unit vector $e_{i^\star} = x^\star$ with $i^\star \in [d_1]$[6] and constant $c_1 := \max_{i \neq i^\star} e_i^\top A y^\star - e_{i^\star}^\top A y^\star > 0$. Then, by definition, we have

$$
\begin{aligned}
x^\top A y^\star - (x^\star)^\top A y^\star &= \sum_{i \neq i^\star} x[i]\left(e_i^\top A y^\star - e_{i^\star} A y^\star\right) \\
&\geq c_1 \sum_{i \neq i^\star} x[i] \\
&= \frac{c_1}{2} \sum_{i \neq i^\star} x[i] + \frac{c_1}{2}(1 - x[i^\star]) \qquad (\textstyle\sum_{i \in [d_1]} x[i] = 1) \\
&\geq \frac{c_1}{2} \sum_{i \neq i^\star} x[i]^2 + \frac{c_1}{2}(1 - x[i^\star])^2 \qquad (x[i] \in [0, 1]) \\
&= \frac{c_1}{2}\|x - x^\star\|^2.
\end{aligned}
$$

Similarly, denote $e_{j^\star} = x^\star$ with $j^\star \in [d_2]$ and constant $c_2 = (x^\star)^\top A y^\star - \max_{j \neq j^\star}(x^\star)^\top A e_j$, we get $(x^\star)^\top A y^\star - \max_{j \neq j^\star}(x^\star)^\top A y \geq \frac{c_2}{2}\|y - y^\star\|^2$. Combining the two inequalities above and let $c = \frac{1}{2}\min(c_1, c_2)$, we have $x^\top A y^\star - (x^\star)^\top A y \geq c\|(x, y) - (x^\star, y^\star)\|^2$ for all $(x, y)$. $\qquad\square$

## F PROOFS FOR SECTION 4

### F.1 PROOFS FOR SECTION 4

*Proof of Lemma 1.* Let $z^\star = (x^\star, y^\star) \in \Delta^{d_1} \times \Delta^{d_2}$ be a Nash equilibrium of the matrix game. For any $z = (Rx, Qy) \in \mathcal{Z}_\geq$, using $\langle F(z), z\rangle = 0$ and the definition of Nash equilibrium, we have $\langle F(z), z - az^\star\rangle = -\langle F(z), az^\star\rangle = a(x^\top A y^\star - (x^\star)^\top A y) \geq 0$. $\qquad\square$

The proof of lemma 3 closely follows that of Theorem 12.1.11 in Facchinei and Pang (2003)

*Proof of Lemma 3.* From Lemma 2, we know

$$(1 - \eta^2 L_F^2) \sum_{t=0}^{\infty} \left\|z^{t+\frac{1}{2}} - z^t\right\|^2 \leq \left\|z^0 - z^\star\right\|^2.$$

---

[6]Given a positive integer $d$, we denote the set $\{1, 2, \ldots, d\}$ by $[d]$.

Since $1 - \eta^2 L_F^2 > 0$, this implies

$$\lim_{t\to\infty} \left\| z^{t+\frac{1}{2}} - z^t \right\| = 0.$$

Using Proposition 4, we know that $\|z^{t+1} - z^{t+\frac{1}{2}}\| \leq \eta L_F \|z^{t+\frac{1}{2}} - z^t\|$. Thus

$$\lim_{t\to\infty} \left\| z^{t+1} - z^t \right\| \leq \lim_{t\to\infty} \left\| z^{t+1} - z^{t+\frac{1}{2}} \right\| + \left\| z^{t+\frac{1}{2}} - z^t \right\| = \lim_{t\to\infty} (1 + \eta L_F) \left\| z^{t+\frac{1}{2}} - z^t \right\| = 0.$$

If $\hat{z}$ is the limit point of the subsequence $\{z^t : t \in \kappa\}$, then we must have

$$\lim_{t(\in\kappa)\to\infty} z^{t+\frac{1}{2}} = \hat{z}.$$

By the definition of $z^{t+\frac{1}{2}}$ in Algorithm 3 and by the continuity of $F$ and of the projection, we get

$$\hat{z} = \lim_{t(\in\kappa)\to\infty} z^{t+\frac{1}{2}} = \lim_{t(\in\kappa)\to\infty} \Pi_{\mathcal{Z}_{\geq}} \left[ z^t - \eta F(z^t) \right] = \Pi_{\mathcal{Z}_{\geq}} [\hat{z} - \eta F(\hat{z})].$$

This shows that $\hat{z} \in SOL(\mathcal{Z}_{\geq}, F)$.

For the third claim, suppose for the sake of contradiction that the sequence $\{\mathrm{DualityGap}(g(z^t))\}$ does not converge to 0. Then we know that there exists a subsequence $\{z^t : t \in \kappa\}$ that converges to $\hat{z}$ and $\mathrm{DualityGap}(g(\hat{z})) > 0$. But since $\hat{z}$ is a limit point of $\{z^t\}$, we have $\hat{z} \in SOL(\mathcal{Z}_{\geq}, F)$. Then $g(\hat{z})$ is a Nash equilibrium and $\mathrm{DualityGap}(g(\hat{z})) = 0$. This gives a contradiction and we conclude that $\lim_{t\to\infty} \mathrm{DualityGap}(g(z^t)) = 0$. □

### F.2 Proofs for Section 4.1

*Proof of Proposition 1.* Denote by $\{z_t\}_{t\in\kappa}$ a subsequence of $\{z^t\}$ that converges to $\hat{z}$. By Lemma 1, the Minty condition holds for $\hat{z}$, so by Lemma 2, $\{\|z^t - \hat{z}\|\}$ is monotonically decreasing and therefore converges. Since $\lim_{t\to\infty} \|z^t - \hat{z}\| = \lim_{t(\in\kappa)\to\infty} \|z^t - \hat{z}\| = 0$, $\{z^t\}$ converges to $\hat{z}$. □

*Proof of Lemma 4.* Let $\{k_i \in \mathbb{Z}\}$ be an increasing sequence of indices such that $\{z^{k_i}\}$ converges to $\hat{z}$ and $\{l_i \in \mathbb{Z}\}$ be an increasing sequence of indices such that $\{z^{l_i}\}$ converges to $\tilde{z}$. Let $\sigma : \{k_i\} \to \{l_i\}$ be a mapping that always maps $k_i$ to a larger index in $\{l_i\}$, i.e. $\sigma(k_i) > k_i$. Such a mapping clearly exists. Since Lemma 2 applies to $az^{\star}$ for any $a \geq 1$, we get

$$\|az^{\star} - \hat{z}\|^2 = \lim_{i\to\infty} \left\| az^{\star} - z^{k_i} \right\|^2 \geq \lim_{i\to\infty} \left\| az^{\star} - z^{\sigma(k_i)} \right\|^2 = \|az^{\star} - \tilde{z}\|^2.$$

By symmetry, the other direction also holds. Thus $\|az^{\star} - \hat{z}\|^2 = \|az^{\star} - \tilde{z}\|^2$ for all $a \geq 1$.

Expanding $\|az^{\star} - \hat{z}\|^2 = \|az^{\star} - \tilde{z}\|^2$ gives

$$\|\hat{z}\|^2 - \|\tilde{z}\|^2 = 2a\langle z^{\star}, \hat{z} - \tilde{z}\rangle, \quad \forall a \geq 1.$$

It implies that $\|\hat{z}\|^2 = \|\tilde{z}\|^2$ and $\langle z^{\star}, \hat{z} - \tilde{z}\rangle = 0$. □

### F.3 Proofs for Section 4.2

We will need the following lemma. Recall the normalization operator $g : \mathbb{R}_+^{d_1} \times \mathbb{R}_+^{d_2} \to \mathcal{Z}$ such that for $z = (z_1, z_2) \in \mathbb{R}_+^{d_1} \times \mathbb{R}_+^{d_2}$, we have $g(z) = (z_1/\|z_1\|_1, z_2/\|z_2\|_1) \in \mathcal{Z}$.

**Lemma 11.** *Let $z \in \Delta^{d_1} \times \Delta^{d_2}$ and $z_1, z_2, z_3 \in \mathcal{Z}_{\geq}$ such that $z_3 = \Pi_{\mathcal{Z}_{\geq}}[z_1 - \eta F(z_2)]$. Denote $(x_3, y_3) = g(z_3)$, then*

$$\mathrm{DualityGap}(x_3, y_3) := \max_{y\in\Delta^{d_2}} x_3^{\top} Ay - \min_{x\in\Delta^{d_1}} x^{\top} Ay_3 \leq \frac{(\|z_1 - z_3\| + \eta L_F \|z_3 - z_2\|)(\|z_3 - z\| + 2)}{\eta}).$$

*Proof of Lemma 11.* Let $\hat{x} \in \arg\min_x x^\top A y_3$ and $\hat{y} \in \arg\max_y x_3^\top A y$. Then we have

$$
\begin{aligned}
&\max_y x_3^\top A y - \min_x x^\top A y_3 \\
&= x_3^\top A \hat{y} - \hat{x}^\top A y_3 \\
&= -\langle F(z_3), \hat{z} \rangle \\
&= \frac{1}{\eta}(\langle z_1 - z_3 + \eta F(z_3) - \eta F(z_2), z_3 - \hat{z} \rangle + \langle z_1 - \eta F(z_2) - z_3, \hat{z} - z_3 \rangle) \\
&\leq \frac{1}{\eta}\langle z_1 - z_3 + \eta F(z_3) - \eta F(z_2), z_3 - \hat{z} \rangle \qquad\qquad (z_3 = \Pi_{\mathcal{Z}_\geq}[z_1 - \eta F(z_2)]) \\
&\leq \frac{(\|z_1 - z_3\| + \eta L_F \|z_3 - z_2\|)\|z_3 - \hat{z}\|}{\eta}.
\end{aligned}
$$

Moreover, we have

$$
\|z_3 - \hat{z}\| \leq \|z_3 - z\| + \|z - \hat{z}\| \leq \|z_3 - z\| + 2.
$$

Combining the above two inequalities completes the proof. $\qquad\square$

We now show a few useful properties of the iterates.

**Proposition 4.** *In the same setup of Lemma 2, for any $t \geq 1$, the following holds.*

1. $\|z^{t+1} - z^{t+\frac{1}{2}}\| \leq \eta L_F \|z^{t+\frac{1}{2}} - z^t\| \leq \|z^{t+\frac{1}{2}} - z^t\|$;

2. $\|z^{t+1} - z^t\| \leq (1 + \eta L_F)\|z^{t+\frac{1}{2}} - z^t\| \leq 2\|z^{t+\frac{1}{2}} - z^t\|$.

*Proof of Proposition 4.* Using the update rule of ExRM$^+$ and the non-expansiveness of the projection operator $\Pi_{\mathcal{Z}_\geq}[\cdot]$ we have

$$
\begin{aligned}
\left\| z^{t+1} - z^{t+\frac{1}{2}} \right\| &\leq \left\| z^t - \eta F(z^t) - (z^t - \eta F(z^{t+\frac{1}{2}})) \right\| \\
&= \eta \left\| F(z^{t+\frac{1}{2}}) - F(z^t) \right\| \leq \eta L_F \left\| z^{t+\frac{1}{2}} - z^t \right\|.
\end{aligned}
$$

Using triangle inequality, we have

$$
\left\| z^{t+1} - z^t \right\| \leq \left\| z^{t+1} - z^{t+\frac{1}{2}} \right\| + \left\| z^{t+\frac{1}{2}} - z^t \right\| \leq (1 + \eta L_F)\left\| z^{t+\frac{1}{2}} - z^t \right\|. \qquad\square
$$

We are now ready to prove Lemma 6.

*Proof of Lemma 6.* Let $z^\star$ be any Nash equilibrium of the game. Since $z^{t+1} = \Pi_{\mathcal{Z}_\geq}[z^t - \eta F(z^{t+\frac{1}{2}})]$, applying Lemma 11 gives

$$
\begin{aligned}
\text{DualityGap}(x^{t+1}, y^{t+1}) &\leq \frac{(\|z^{t+1} - z^t\| + \eta L_F \|z^{t+1} - z^{t+\frac{1}{2}}\|)(\|z^{t+1} - z^\star\| + 2)}{\eta} \\
&\leq \frac{3\|z^{t+\frac{1}{2}} - z^t\|(\|z^{t+1} - z^\star\| + 2)}{\eta} \qquad (\eta L_F \leq 1 \text{ and Proposition } 4) \\
&\leq \frac{3\|z^{t+\frac{1}{2}} - z^t\|(\|z^0 - z^\star\| + 2)}{\eta} \qquad\qquad\quad (\text{Lemma } 2) \\
&\leq \frac{12\|z^{t+\frac{1}{2}} - z^t\|}{\eta}, \qquad (z^0 \text{ and } z^\star \text{ are both in the simplex } \Delta^{d_1} \times \Delta^{d_2})
\end{aligned}
$$

which finishes the proof. $\qquad\square$

### F.4 PROOFS FOR SECTION 4.3

*Proof of Theorem 4.* We note that RS-ExRM$^+$ always restarts in the first iteration since $\|z^{\frac{1}{2}} - z^0\| \le \frac{\|z^0 - z^\star\|}{\sqrt{1-\eta^2 L_F^2}} \le \frac{2}{\sqrt{1-\eta^2 L_F^2}}$. Let $1 = t_1 < t_2 < \ldots$ be the iterations where the algorithm restarts with $(x^{t_k}, y^{t_k})$. Using Lemma 6 and the restart condition, we know that for any $k \ge 1$,

$$\mathrm{DualityGap}(x^{t_k}, y^{t_k}) \le \frac{12\|z^{t_k - \frac{1}{2}} - z^{t_k - 1}\|}{\eta} \le \frac{48}{\eta\sqrt{1-\eta^2 L_F^2}} \cdot \frac{1}{2^k}$$

$$\left\|(x^{t_k}, y^{t_k}) - \Pi_{\mathcal{Z}^\star}[(x^{t_k}, y^{t_k})]\right\| \le \frac{\mathrm{DualityGap}(x^{t_k}, y^{t_k})}{c} \le \frac{48}{c\eta\sqrt{1-\eta^2 L_F^2}} \cdot \frac{1}{2^k}.$$

Moreover, for any iterate $t \in [t_k + 1, t_{k+1} - 1]$, using Lemma 6, we get

$$\mathrm{DualityGap}(x^t, y^t) \le \frac{12\|z^{t-\frac{1}{2}} - z^{t-1}\|}{\eta} \le \frac{12\|z^{t_k} - \Pi_{\mathcal{Z}^\star}[(x^{t_k}, y^{t_k})]\|}{\eta\sqrt{1-\eta^2 L_F^2}} \le \frac{576}{c\eta^2(1-\eta^2 L_F^2)} \cdot \frac{1}{2^k}.$$

Using metric subregularity (Proposition 2), we have for any iterate $t \in [t_k + 1, t_{k+1} - 1]$,

$$\left\|(x^t, y^t) - \Pi_{\mathcal{Z}^\star}[(x^t, y^t)]\right\| \le \frac{576}{c^2\eta^2(1-\eta^2 L_F^2)} \cdot \frac{1}{2^k}.$$

If we can prove $t_{k+1} - t_k$ is always bounded by a constant $C \ge 1$, then $t_k \le C(k-1) + 1 \le Ck$. Then for any $t \ge 1$, we have $t \ge t_{\lfloor \frac{t}{C} \rfloor}$ and

$$\left\|(x^t, y^t) - \Pi_{\mathcal{Z}^\star}[(x^t, y^t)]\right\| \le \frac{576}{c^2\eta^2(1-\eta^2 L_F^2)} \cdot \frac{1}{2^{\lfloor \frac{t}{C} \rfloor}} \le \frac{576}{c^2\eta^2(1-\eta^2 L_F^2)} \cdot 2^{-\frac{t}{C}} \le \frac{576}{c^2\eta^2(1-\eta^2 L_F^2)} \cdot (1 - \frac{1}{3C})^t.$$

**Bounding $t_{k+1} - t_k$** If $t_{k+1} - t_k = 1$, then the claim holds. Now we assume $t_{k+1} - t_k \ge 2$. Before restarting, the updates of $z^t$ for $t_k \le t \le t_{k+1} - 2$ is just ExRM$^+$. By Lemma 2, we know

$$\sum_{t=t_k}^{t_{k+1}-2} \left\|z^{t+\frac{1}{2}} - z^t\right\|^2 \le \frac{\|(x^{t_k}, y^{t_k}) - \Pi_{\mathcal{Z}^\star}[(x^{t_k}, y^{t_k})]\|^2}{1 - \eta^2 L_F^2}$$

It implies that there exists $t \in [t_k, t_{k+1} - 2]$ such that

$$\left\|z^{t+\frac{1}{2}} - z^t\right\| \le \frac{\|(x^{t_k}, y^{t_k}) - \Pi_{\mathcal{Z}^\star}[(x^{t_k}, y^{t_k})]\|}{\sqrt{1-\eta^2 L_F^2}\sqrt{t_{k+1} - t_k - 1}} \le \frac{48}{c\eta(1-\eta^2 L_F^2)\sqrt{t_{k+1} - t_k - 1} \cdot 2^k}.$$

On the other hand, since the algorithm does not restart in $[t_k, t_{k+1} - 2]$, we know for every $t \in [t_k, t_{k+1} - 2]$,

$$\left\|z^{t+\frac{1}{2}} - z^t\right\| > \frac{4}{\sqrt{1-\eta^2 L_F^2} \cdot 2^{k+1}}.$$

Combining the above inequalities gives

$$t_{k+1} - t_k - 1 \le \frac{48^2}{c^2\eta^2(1-\eta^2 L_F^2)^2 \cdot 2^{2k}} \cdot \frac{(1-\eta^2 L_F^2) \cdot 2^{2k+2}}{16}$$

$$= \frac{48^2/4}{c^2\eta^2(1-\eta^2 L_F^2)} = \frac{576}{c^2\eta^2(1-\eta^2 L_F^2)}.$$

**Linear Last-Iterate Convergence** Combing all the above, we get for all $t \ge 1$,

$$\mathrm{DualityGap}(x^t, y^t) \le \frac{576}{c\eta^2(1-\eta^2 L^2)} \cdot \left(1 - \frac{1}{3(1 + \frac{576}{c^2\eta^2(1-\eta^2 L_F^2)})}\right)^t$$

$$\left\|(x^t, y^t) - \Pi_{\mathcal{Z}^\star}[(x^t, y^t)]\right\| \le \frac{576}{c^2\eta^2(1-\eta^2 L^2)} \cdot \left(1 - \frac{1}{3(1 + \frac{576}{c^2\eta^2(1-\eta^2 L_F^2)})}\right)^t.$$

This completes the proof. □

## G PROOFS FOR SECTION 5

### G.1 ASYMPTOTIC CONVERGENCE OF SPRM$^+$

In the following, we will prove last-iterate convergence of SPRM$^+$ (Algorithm 4). The proof follows the same idea as that of ExRM$^+$ in previous sections. Applying standard analysis of OG, we have the following important lemma for SPRM$^+$.

**Lemma 12** (Adapted from Lemma 1 in (Wei et al., 2021)). *Let $z^\star \in \mathcal{Z}_\geq$ be a point such that $\langle F(z), z - z^\star \rangle \geq 0$ for all $z \in \mathcal{Z}_\geq$. Let $\{z^t\}$ and $\{w^t\}$ be the sequences produced by SPRM$^+$. Then for every iteration $t \geq 0$ it holds that*

$$\left\| w^{t+1} - z^\star \right\|^2 + \frac{1}{16} \left\| w^{t+1} - z^t \right\|^2 \leq \left\| w^t - z^\star \right\|^2 + \frac{1}{16} \left\| w^t - z^{t-1} \right\|^2 - \frac{15}{16} \left( \left\| w^{t+1} - z^t \right\|^2 + \left\| w^t - z^t \right\|^2 \right).$$

*It also implies for any $t \geq 0$,*

$$\left\| w^{t+1} - z^t \right\| + \left\| w^t - z^t \right\| \leq 2 \left\| w^0 - z^\star \right\|, \quad \left\| w^t - z^t \right\| + \left\| w^t - z^{t-1} \right\| \leq 2 \left\| w^0 - z^\star \right\|.$$

**Lemma 13.** *Let $\{z^t\}$ and $\{w^t\}$ be the sequences produced by SPRM$^+$, then*

1. *The sequences $\{w^t\}$ and $\{z^t\}$ are bounded and thus have at least one limit point.*

2. *If the sequence $\{w^t\}$ converges to $\hat{w} \in \mathcal{Z}_\geq$, then the sequence $\{z^t\}$ also converges to $\hat{w}$.*

3. *If $\hat{w}$ is a limit point of $\{w^t\}$, then $\hat{w} \in SOL(\mathcal{Z}_\geq, F)$ and $g(\hat{w})$ is a Nash equilibrium of the game.*

*Proof.* By Lemma 12 and the fact that $w^0 = z^{-1}$, we know for any $t \geq 0$,

$$\left\| w^{t+1} - z^\star \right\|^2 + \frac{1}{16} \left\| w^{t+1} - z^t \right\|^2 \leq \left\| w^0 - z^\star \right\|^2,$$

which implies the sequences $\{w^t\}$ and $\{z^t\}$ are bounded. Thus, both of them have at least one limit point. By Lemma 12, we have

$$\frac{15}{32} \sum_{t=1}^T \left\| w^{t+1} - w^t \right\|^2 \leq \frac{15}{16} \sum_{t=1}^T \left( \left\| w^{t+1} - z^t \right\|^2 + \left\| w^t - z^t \right\|^2 \right) \leq \left\| w^0 - z^\star \right\|^2 + \frac{1}{16} \left\| w^0 - z^{-1} \right\|^2.$$

This implies

$$\lim_{t \to \infty} \left\| w^{t+1} - w^t \right\| = 0, \quad \lim_{t \to \infty} \left\| w^t - z^t \right\| = 0, \quad \lim_{t \to \infty} \left\| w^{t+1} - z^t \right\| = 0.$$

If $\hat{w}$ is the limit point of the subsequence $\{w^t : t \in \kappa\}$, then we must have

$$\lim_{(t \in \kappa) \to \infty} w^{t+1} = \hat{w}, \quad \lim_{(t \in \kappa) \to \infty} z^t = \hat{w}.$$

By the definition of $w^{t+1}$ in SPRM$^+$ and by the continuity of $F$ and of the projection, we get

$$\hat{w} = \lim_{t (\in \kappa) \to \infty} w^{t+1} = \lim_{t (\in \kappa) \to \infty} \Pi_{\mathcal{Z}_\geq} \left[ w^t - \eta F(z^t) \right] = \Pi_{\mathcal{Z}_\geq} [\hat{w} - \eta F(\hat{w})].$$

This shows that $\hat{w} \in SOL(\mathcal{Z}_\geq, F)$. By Lemma 11, we know $g(\hat{w})$ is a Nash equilibrium of the game. $\qquad \square$

**Proposition 5.** *Let $\{z^t\}$ and $\{w^t\}$ be the sequences produced by SPRM$^+$. If $\{w^t\}$ has a limit point $\hat{w}$ such that $\hat{w} = az^\star$ for $z^\star \in \Delta^{d_1} \times \Delta^{d_2}$ and $a \geq 1$ (equivalently, colinear with a pair of strategies in the simplex), then $\{w^t\}$ converges to $\hat{w}$.*

*Proof.* By Lemma 1, we know the MVI condition holds for $\hat{w}$, Then by Lemma 12, $\{\|w^t - \hat{w}\|^2 + \frac{1}{16} \|w^t - z^{t-1}\|^2\}$ is monotonitically decreasing and therefore converges. Let $\hat{w}$ be the limit of the sequence $\{w^t : t \in \kappa\}$. Using the fact that $\lim_{t \to \infty} \|w^t - z^{t-1}\| = 0$, we know that $\{\|w^t - \hat{w}\|\}$ also converges and

$$\lim_{t \to \infty} \left\| w^t - \hat{w} \right\|^2 + \frac{1}{16} \left\| w^t - z^{t-1} \right\|^2 = \lim_{t \to \infty} \left\| w^t - \hat{w} \right\|^2 = \lim_{t (\in \kappa) \to \infty} \left\| w^t - \hat{w} \right\|^2 = 0$$

Thus, $\{w^t\}$ converges to $\hat{w}$. $\qquad \square$

**Lemma 14** (Structure of Limit Points)**.** *Let $\{z^t\}$ and $\{w^t\}$ be the sequences produced by SPRM$^+$. Let $z^\star \in \Delta^{d_1} \times \Delta^{d_2}$ be any Nash equilibrium of $A$. If $\hat{w}$ and $\tilde{w}$ are two limit points of $\{w^t\}$, then the following holds.*

1. *$\|az^\star - \hat{w}\|^2 = \|az^\star - \tilde{w}\|^2$ for all $a \geq 1$.*

2. *$\|\hat{w}\|^2 = \|\tilde{w}\|^2$.*

3. *$\langle z^\star, \hat{w} - \tilde{w} \rangle = 0$.*

*Proof.* Let $\{k_i \in \mathbb{Z}\}$ be an increasing sequence of indices such that $\{w^{k_i}\}$ converges to $\hat{w}$ and $\{l_i \in \mathbb{Z}\}$ be an increasing sequence of indices such that $\{w^{l_i}\}$ converges to $\tilde{w}$. Let $\sigma : \{k_i\} \to \{l_i\}$ be a mapping that always maps $k_i$ to a larger index in $\{l_i\}$, i.e. $\sigma(k_i) > k_i$. Such a mapping clearly exists. Since Lemma 12 applies to $az^\star$ for any $a \geq 1$ and $\lim_{t \to \infty} \|w^t - z^{t-1}\| = 0$, we get

$$
\|az^\star - \hat{w}\|^2 = \lim_{i \to \infty} \|az^\star - w^{k_i}\|^2 + \frac{1}{16}\|w^{k_i} - z^{k_i - 1}\|^2
$$

$$
\geq \lim_{i \to \infty} \left\|az^\star - w^{\sigma(k_i)}\right\|^2 + \frac{1}{16}\left\|w^{\sigma(k_i)} - z^{\sigma(k_i) - 1}\right\|^2
$$

$$
= \|az^\star - \tilde{w}\|^2.
$$

By symmetry, the other direction also holds. Thus, $\|az^\star - \hat{w}\|^2 = \|az^\star - \tilde{w}\|^2$ for all $a \geq 1$.

Expanding $\|az^\star - \hat{w}\|^2 = \|az^\star - \tilde{w}\|^2$ gives

$$
\|\hat{w}\|^2 - \|\tilde{w}\|^2 = 2a\langle z^\star, \hat{w} - \tilde{w} \rangle, \quad \forall a \geq 1.
$$

It implies that $\|\hat{w}\|^2 = \|\tilde{w}\|^2$ and $\langle z^\star, \hat{w} - \tilde{w} \rangle = 0$. □

With Lemma 12, Lemma 13, Proposition 5, and Lemma 14, by the same argument as the proof for ExRM$^+$ in Lemma 5, we can prove that $\{w^t\}$ produced by SPRM$^+$ has a unique limit point and thus converges, as shown in the following lemma.

**Lemma 15** (Uniqueness of Limit Point)**.** *The iterates $\{w^t\}$ produced by SPRM$^+$ have a unique limit point.*

Thus, Theorem 5 directly follows from Lemma 13 and Lemma 15.

## G.2 BEST-ITERATE CONVERGENCE OF SPRM$^+$

**Lemma 16.** *Let $\{z^t\}$ and $\{w^t\}$ be the sequences produced by SPRM$^+$ (Algorithm 4). Then*

1. $\text{DualityGap}(g(w^{t+1})) \leq \frac{5(\|w^t - z^t\| + \|w^{t+1} - z^t\|)}{\eta}$;

2. $\text{DualityGap}(g(z^t)) \leq \frac{9(\|w^t - z^t\| + \|w^t - z^{t-1}\|)}{\eta}$.

.

*Proof of Lemma 16.* Since $w^{t+1} = \Pi_{\mathcal{Z}_\geq}[w^t - \eta F(z^t)]$, by Lemma 11, we know for any $z^\star \in \mathcal{Z}^\star$

$$
\text{DualityGap}(g(w^{t+1})) \leq \frac{(\|w^{t+1} - w^t\| + \eta L_F \|w^{t+1} - z^t\|)(\|w^{t+1} - z^\star\| + 2)}{\eta}
$$

$$
\leq \frac{(1 + \eta L_F)(\|w^t - z^t\| + \|w^{t+1} - z^t\|)(\|w^{t+1} - z^\star\| + 2)}{\eta}
$$

$$
\leq \frac{(1 + \eta L_F)(\|w^t - z^t\| + \|w^{t+1} - z^t\|)(\|w^0 - z^\star\| + 2)}{\eta}
$$

$$
\leq \frac{5(\|w^t - z^t\| + \|w^{t+1} - z^t\|)}{\eta},
$$

where in the second inequality we apply $\|w^{t+1} - w^t\| \leq \|w^{t+1} - z^t\| + \|w^t - z^t\|$; in the third inequality we use $\|w^{t+1} - z^\star\| \leq \|w^0 - z^\star\|$ by Lemma 12; in the last inequality we use $\eta L_F \leq \frac{1}{8}$ and $\|w^0 - z^\star\| \leq 2$ since $w^0, z^\star \in \Delta^{d_1} \times \Delta^{d_2}$.

Similarly, since $z^t = \Pi_{\mathcal{Z}_\geq}[w^t - \eta F(z^{t-1})]$, by Lemma 11, we know for any $z^\star \in \mathcal{Z}^\star$

$$\text{DualityGap}(g(z^t))$$
$$\leq \frac{(\|z^t - w^t\| + \eta L_F \|z^t - z^{t-1}\|)(\|z^t - z^\star\| + 2)}{\eta}$$
$$\leq \frac{(1 + \eta L_F)(\|w^t - z^t\| + \|w^t - z^{t-1}\|)(\|w^t - z^\star\| + \|w^t - z^t\| + 2)}{\eta}$$
$$\leq \frac{(1 + \eta L_F)(\|w^t - z^t\| + \|w^t - z^{t-1}\|)(\|w^0 - z^\star\| + 2\|w^0 - z^\star\| + 2)}{\eta}$$
$$\leq \frac{9(\|w^t - z^t\| + \|w^t - z^{t-1}\|)}{\eta},$$

where in the second inequality we apply $\|z^t - z^{t-1}\| \leq \|z^t - w^t\| + \|w^t - z^{t-1}\|$ and $\|z^t - z^\star\| \leq \|z^t - w^t\| + \|w^t - z^\star\|$; in the third inequality we use $\|w^{t+1} - z^\star\| \leq \|w^0 - z^\star\|$ and $\|w^t - z^t\| \leq 2\|w^0 - z^\star\|$ by Lemma 12; in the last inequality we use $\eta L_F \leq \frac{1}{8}$ and $\|w^0 - z^\star\| \leq 2$ since $w^0, z^\star \in \Delta^{d_1} \times \Delta^{d_2}$. □

*Proof of Theorem 6.* Fix any Nash equilibrium $z^\star$ of the game. From Lemma 12, we know that

$$\sum_{t=0}^{T-1} \left( \|w^{t+1} - z^t\|^2 + \|w^t - z^t\|^2 \right) \leq \frac{16}{15} \cdot \|w^0 - z^\star\|^2.$$

This implies that there exists $0 \leq t \leq T - 1$ such that $\|w^{t+1} - z^t\| + \|w^t - z^t\| \leq \frac{2\|w^0 - z^\star\|}{\sqrt{T}}$ (we use $a + b \leq \sqrt{2(a^2 + b^2)}$). By applying Lemma 16, we then get

$$\text{DualityGap}(g(w^{t+1})) \leq \frac{5(\|w^t - z^t\| + \|w^{t+1} - z^t\|)}{\eta} \leq \frac{10\|w^0 - z^*\|}{\eta\sqrt{T}}.$$

Similarly, using Lemma 12 and the fact that $w^0 = z^{-1}$, we know

$$\sum_{t=0}^{T} \left( \|w^t - z^t\|^2 + \|w^t - z^{t-1}\|^2 \right) \leq \sum_{t=0}^{T} \left( \|w^t - z^t\|^2 + \|w^{t+1} - z^t\|^2 \right) \leq \frac{16}{15} \cdot \|w^0 - z^\star\|^2.$$

This implies that there exists $0 \leq t \leq T$ such that $\|w^t - z^t\| + \|w^t - z^{t-1}\| \leq \frac{2\|w^0 - z^\star\|}{\sqrt{T}}$ (we use $a + b \leq \sqrt{2(a^2 + b^2)}$). By applying Lemma 16, we then get

$$\text{DualityGap}(g(z^t)) \leq \frac{9(\|w^t - z^t\| + \|w^t - z^{t-1}\|)}{\eta} \leq \frac{18\|w^0 - z^*\|}{\eta\sqrt{T}}.$$

□

### G.3 LINEAR CONVERGENCE OF RS-SPRM⁺.

*Proof of Theorem 7.* We note that Algorithm 6 always restarts in the first iteration $t = 1$ since $\|w^1 - z^0\| + \|w^0 - z^0\| \leq \sqrt{\frac{32}{15}\|w^0 - z^\star\|^2} \leq 4 = \frac{8}{2^1}$. We denote $1 = t_1 < t_2 < \ldots < t_k < \ldots$ the iteration where Algorithm 6 restarts for the $k$-th time, i.e., the "If" condition holds. Then by Lemma 16, we know for every $t_k$, $w^{t_k} = g(w^{t_k}) \in \mathcal{Z} = \Delta^{d_1} \times \Delta^{d_2}$ and

$$\text{DualityGap}(w^{t_k}) \leq \frac{5(\|w^{t_k} - z^{t_k-1}\| + \|w^{t_k-1} - z^{t_k-1}\|)}{\eta} \leq \frac{40}{\eta \cdot 2^k},$$

and by metric subregularity (Proposition 2), we have

$$\left\| w^{t_k} - \Pi_{\mathcal{Z}^\star}[w^{t_k}] \right\| \le \frac{\mathrm{DualityGap}(g(w^{t_k}))}{c} \le \frac{40}{c\eta \cdot 2^k}.$$

Then, for any iteration $t_k + 1 \le t \le t_{k+1} - 1$, using Lemma 12 and Lemma 16, we have

$$\mathrm{DualityGap}(g(w^t)) \le \frac{5(\|w^t - z^{t-1}\| + \|w^{t-1} - z^{t-1}\|)}{\eta} \qquad \text{(Lemma 16)}$$

$$\le \frac{10\|w^{t_k} - \Pi_{\mathcal{Z}^\star}[w^{t_k}]\|}{\eta} \qquad \text{(Lemma 12)}$$

$$\le \frac{400}{c\eta^2 \cdot 2^k}.$$

Then again by metric subregularity, we also get $\|g(w^t) - \Pi_{\mathcal{Z}^\star}[g(w^t)]\| \le \frac{400}{c^2\eta^2 \cdot 2^k}$ for every $t \in [t_k + 1, t_{k+1} - 1]$.

Similarly, for any iteration $t_k \le t \le t_{k+1} - 1$, using Lemma 12 and Lemma 16, we have

$$\mathrm{DualityGap}(g(z^t)) \le \frac{9(\|w^t - z^t\| + \|w^t - z^{t-1}\|)}{\eta} \qquad \text{(Lemma 16)}$$

$$\le \frac{18\|w^{t_k} - \Pi_{\mathcal{Z}^\star}[w^{t_k}]\|}{\eta} \qquad \text{(Lemma 12)}$$

$$\le \frac{720}{c\eta^2 \cdot 2^k}.$$

Then by metric subregularity, we also get $\|g(z^t) - \Pi_{\mathcal{Z}^\star}[g(z^t)]\| \le \frac{400}{c^2\eta^2 \cdot 2^k}$ for every $t \in [t_k, t_{k+1} - 1]$.

**Bounding $t_{k+1} - t_k$**   Fix any $k \ge 1$. If $t_{k+1} = t_k + 1$, then we are good. Now we assume $t_{k+1} > t_k + 1$. By Lemma 12, we know

$$\sum_{t=t_k}^{t_{k+1}-2} \left( \left\| w^{t+1} - z^t \right\|^2 + \left\| w^t - z^t \right\|^2 \right) \le \frac{16}{15} \left\| w^{t_k} - \Pi_{\mathcal{Z}^\star}[w^{t_k}] \right\|^2.$$

This implies that there exists $t \in [t_k, t_{k+1} - 2]$ such that

$$\left\| w^{t+1} - z^t \right\| + \left\| w^t - z^t \right\| \le \frac{2\|w^{t_k} - \Pi_{\mathcal{Z}^\star}[w^{t_k}]\|}{\sqrt{t_{k+1} - t_k - 1}} \le \frac{80}{c\eta 2^k} \frac{1}{\sqrt{t_{k+1} - t_k - 1}}.$$

On the other hand, since Algorithm 6 does not restart in $[t_k, t_{k+1} - 2]$, we have for every $t \in [t_k, t_{k+1} - 2]$,

$$\left\| w^{t+1} - z^t \right\| + \left\| w^t - z^t \right\| > \frac{8}{2^{k+1}}.$$

Combining the above two inequalities, we get

$$t_{k+1} - t_k - 1 \le \frac{400}{c^2\eta^2}.$$

**Linear Last-Iterate Convergence Rates**   Define $C := 1 + \frac{400}{c^2\eta^2} \ge t_{k+1} - t_k$. Then $t_k \le Ck$ and for any $t \ge 1$, we have $t \ge t_{\lfloor \frac{t}{C} \rfloor}$ and

$$\mathrm{DualityGap}(g(w^t)) \le \frac{400}{c\eta^2} \cdot 2^{-\lfloor \frac{t}{C} \rfloor} \le \frac{400}{c\eta^2} \left( 1 - \frac{1}{3C} \right)^t = \frac{400}{c\eta^2} \cdot \left( 1 - \frac{1}{3(1 + \frac{400}{c^2\eta^2})} \right)^t.$$

Using metric subregularity, we get for any $t \ge 1$.

$$\left\| g(w^t) - \Pi_{\mathcal{Z}^\star}[g(w^t)] \right\| \le \frac{400}{c^2\eta^2} \cdot \left( 1 - \frac{1}{3(1 + \frac{400}{c^2\eta^2})} \right)^t.$$

Similarly, we have for any $t \ge 1$

$$\mathrm{DualityGap}(g(z^t)) \le \frac{720}{c\eta^2} \cdot \left( 1 - \frac{1}{3(1 + \frac{400}{c^2\eta^2})} \right)^t, \left\| g(z^t) - \Pi_{\mathcal{Z}^\star}[g(z^t)] \right\| \le \frac{720}{c^2\eta^2} \cdot \left( 1 - \frac{1}{3(1 + \frac{400}{c^2\eta^2})} \right)^t.$$

This completes the proof. $\qquad\square$

# H    ADDITIONAL DETAILS ON NUMERICAL EXPERIMENTS

## H.1    EXTRAGRADIENT ALGORITHM AND OPTIMISTIC GRADIENT

We describe the extragradient algorithm (EG) and optimistic gradient descent (OG) in Algorithm 9 and Algorithm 10. Recall that the gradient operator gradient operator $G : \mathcal{Z} = \Delta^{d_1} \times \Delta^{d_2} \to \mathbb{R}^{d_1} \times \mathbb{R}^{d_2}$ is defined as $G(z) = (Ay, -A^\top x)$ for $z = (x, y)$.

| **Algorithm 9** Extragradient algorithm (EG) | **Algorithm 10** Optimistic gradient descent (OG) |
|---|---|
| 1: **Input**: Step size $\eta > 0$. | 1: **Input**: Step size $\eta > 0$. |
| 2: **Initialize**: $z^0 \in \mathcal{Z}$ | 2: **Initialize**: $z^{-1} = w^0 \in \mathcal{Z}$ |
| 3: **for** $t = 0, 1, \dots$ **do** | 3: **for** $t = 0, 1, \dots$ **do** |
| 4:     $z^{t+1/2} = \Pi_{\mathcal{Z}} \left( z^t - \eta G(z^t) \right)$ | 4:     $z^t = \Pi_{\mathcal{Z}} \left( w^t - \eta G(z^{t-1}) \right)$ |
| 5:     $z^{t+1} = \Pi_{\mathcal{Z}} \left( z^t - \eta G(z^{t+1/2}) \right)$ | 5:     $w^{t+1} = \Pi_{\mathcal{Z}} \left( w^t - \eta G(z^t) \right)$ |

## H.2    GAME INSTANCES

Below we describe the extensive-form benchmark game instances we test in our experiments. These games are solved in their *normal-form* representation. For each game, we report the size of the payoff matrix in this representation. All the algorithms were coded with Python 3.8.8, and we ran our numerical experiments on a laptop with 2.2 GHz Intel Core i7 and 8 GB of RAM. In this setup, he performance of each algorithm on the instances considered in this paper (see details below) can be computed in a few hours.

**Kuhn poker**    Kuhn poker is a widely used benchmark game introduced by Kuhn (1950). At the beginning of the game, each player pays one chip to the pot, and each player is dealt a single private card from a deck containing three cards: jack, queen, and king. The first player can check or bet, i.e., putting an additional chip in the pot. Then, the second player can check or bet after the first player's check, or fold/call the first player's bet. After a bet of the second player, the first player still has to decide whether to fold or to call the bet. At the showdown, the player with the highest card who has not folded wins all the chips in the pot.

The payoff matrix for Kuhn poker has dimension $27 \times 64$ and 690 nonzeros.

**Goofspiel**    We use an imperfect-information variant of the standard benchmark game introduced by Ross (1971). We use a 4-rank variant, that is, each player has a hand of cards with values $\{1, 2, 3, 4\}$. A third stack of cards with values $\{1, 2, 3, 4\}$ (in order) is placed on the table. At each turn, a prize card is revealed, and each player privately chooses one of his/her cards to bid. The players do not reveal the cards that they have selected. Rather, they show their cards to a fair umpire, which determines which player has played the highest card and should receive the prize card as a result. In case of a tie, the prize is split evenly among the winners. After 4 turns, all the prizes have been dealt out and the game terminates. The payoff of each player is computed as the sum of the values of the cards they won.

The payoff matrix for Goofspiel has dimension $72 \times 7,808$ and $562,176$ nonzeros.

**Random instances.**    We consider random matrices of size $(10, 15)$. The coefficients of the matrices are normally distributed with mean $0$ and variance $1$ and are sampled using the `numpy.random.normal` function from the `numpy` Python package (Harris et al., 2020). In all figures, we average the last-iterate duality gaps over the 25 random matrix instances, and we also show the confidence intervals.

## H.3    EXPERIMENTS WITH RESTARTING

In this section, we compare the last-iterate convergence performances of ExRM$^+$, SPRM$^+$ and their restarting variants RS-ExRM$^+$ and RS-SPRM$^+$ as introduced in Section 4.3 and Section 5. We

present our results when choosing a stepsize $\eta = 0.05$ in Figure 7 and Figure 8. To better illustrate the linear convergence rate, we use semi-log plot in Figure 8 and plot a reference line $(1 - 0.002)^t$. The numerical results show that both RS-ExRM$^+$ and RS-SPRM$^+$ have linear last-iterate convergence and confirm our theoretical results in Theorem 4 and Theorem 7. Moreover, we also observe that restarting accelerates the convergence in these two examples.

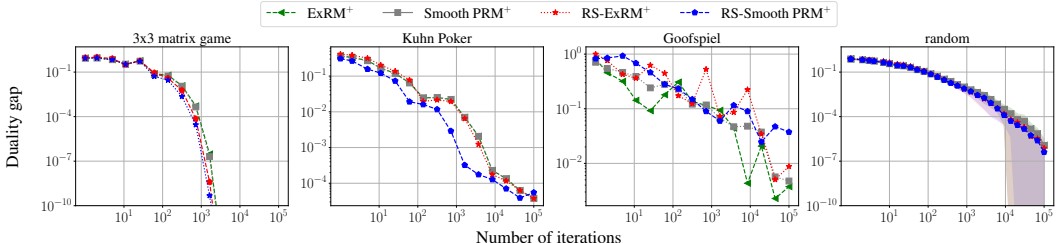

Figure 7: For a stepsize $\eta = 0.05$, empirical performances of several algorithms on our $3 \times 3$ matrix game (left plot), Kuhn Poker and Goofspiel (center plots) and on random instances (right plot).

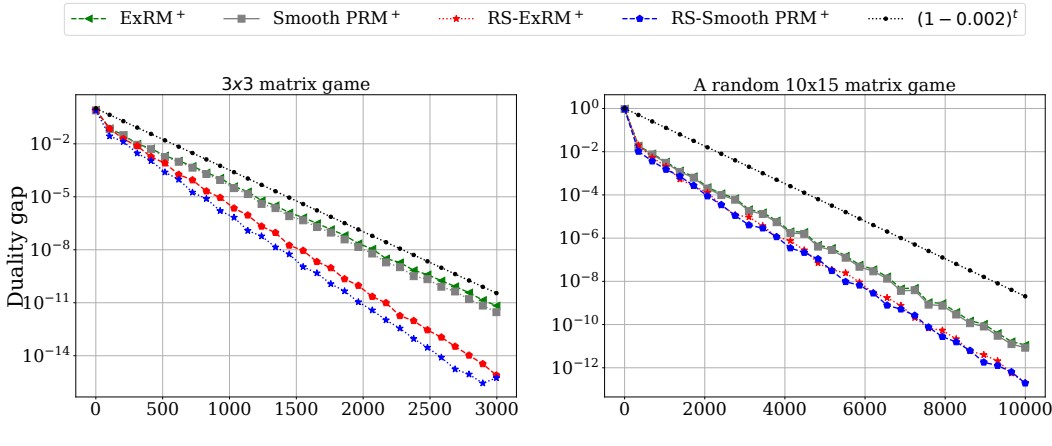

Figure 8: The last-iterate convergence rate of ExRM$^+$ and SPRM$^+$ and their restarting variants RS-ExRM$^+$ and RS-SPRM$^+$ on the $3 \times 3$ hard instance and a random $10 \times 15$ matrix game. The step size is $\eta = 0.05$ for all algorithms. We also plot the line $(1 - 0.002)^t$ for reference.

## H.4 EXPERIMENTS WITH BEST-ITERATE CONVERGENCE

In this section we compare the empirical performances of the best-iterate visited by the algorithms studied in this paper. For the sake of completeness we also show the best-iterate convergence of EG and OG. We present our results in Figure 9, where we also show the line associated with $1/\sqrt{T}$ for reference. The setup is the same as for Figure 3 in Section 6.

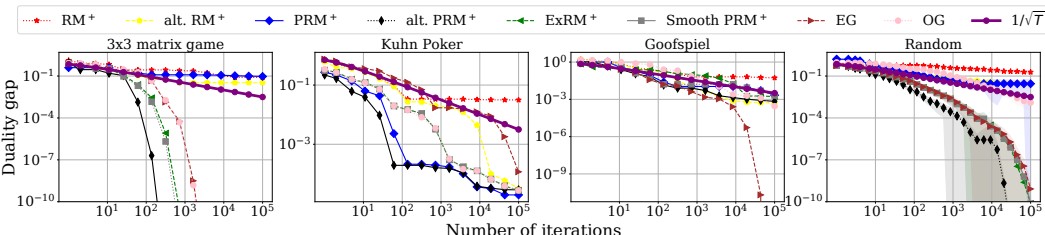

Figure 9: Empirical best-iterate performances of several algorithms on the $3 \times 3$ matrix game (left plot), Kuhn poker and Goofspiel (center plots), and random instances (right plot).

