# OpenReview forum: "Last-Iterate Convergence Properties of Regret-Matching Algorithms in Games"
_ICLR.cc/2025/Conference — ICLR 2025 Poster_

### Official Review · Reviewer_QZ5e · 2024-10-22

**Soundness:** 3
**Presentation:** 3
**Contribution:** 3
**Rating:** 8
**Confidence:** 4

**Summary:**

This paper investigates the last-iterate convergence rate of RM+ algorithms. Firstly, they provide numerical evidence that RM+ and PRM+,may fail to have asymptotic last-iterate convergence and best-iterate convergence. Secondly, they prove that ExRM+ and SPRM+ exhibit asymptotic last-iterate convergence. Lastly, they prove an $O(1/T)$ best-iterate convergence for ExRM+ and SPRM+, and use this best-iterate convergence to introduce new variants of ExRM+ and SPRM+, which achieve linear last-iterate convergence.

**Strengths:**

This paper was previously submitted to ICLR 2024. This paper investigates the convergence of RM+ algorithms, an important topic that has been missing in prior research. It offers valuable insights for future studies. To the best of my knowledge, this is the second paper to explore the convergence properties of RM+ algorithms.

**Weaknesses:**

1. Although this paper considers a different game from [Meng et al., 2023], its analysis of the convergence of RM+ largely depends on the analysis of the convergence of RM+ in [Meng et al., 2023]. More precisely, the proofs of RM+ convergence in this paper and in [Meng et al., 2023] are identical. The primary novelty of convergence results of RM+ of this paper relative to [Meng et al., 2023] lies in the properties of strict NE. Therefore, the authors should revise the introduction to remove the claim that [Meng et al., 2023] is "incomparable to our matrix game setting" and instead explicitly acknowledge that the convergence proof of RM+ of this paper largely depends on [Meng et al., 2023].

2. The second, and most critical, weakness of this paper is the lack of clarity in its explanation of the convergence of the iterates.
   - Convergence in the Duality Gap indicates that the iterates converge to the set of NE. For solving NE, this is sufficient, as the goal is merely to find one NE.
   - The metric used in the first author's previous works about the  last-iterate convergence of OMD-based algorithms, Tangent Residual means the upper bound of Duality Gap. In other words, the convergence in the tangent residual means the convergence in the duality gap. Why the convergence in the tangent residual is this sufficient to claim the last-iterate convergence in the earlier works but not in this paper?
   - In Section 7 Conclusions, the authors claim “linear-rate convergence in iterates.” However, the metric for linear-rate convergence is also the Duality Gap.

**Questions:**

1. What is the exact difference between "Convergence of the iterates" and "Convergence in the Duality Gap"? What are the advantages of "Convergence of the iterates" compared to "Convergence in the Duality Gap"?

2. Why does the first author in prior works use tangent residual to claim "last-iterate convergence," while this paper distinguishes between "Convergence of the iterates" and "Convergence in the Duality Gap"?

3. Is the claim in the conclusion about "linear-rate convergence in iterates" correct?

If the authors can address my concerns regarding the "Convergence of the iterates," I would be willing to increase my score.

---

> ### Author Response · Authors · 2024-11-18
>
> Thank you for the positive review and comments! We address your comments and questions below.
>
> >  Although this paper considers a different game from [Meng et al., 2023] ... instead explicitly acknowledge that the convergence proof of RM+ of this paper largely depends on [Meng et al., 2023].
>
> A: We have updated the introduction and added a discussion with [Meng et al., 2023] as follows: “[Meng et al., 2023] studies RM+ in strongly-convex-strongly-concave games and proves its asymptotic convergence (without rate) to the unique equilibrium. Theorem 1’s proof largely follows from [Meng et al., 2023] with the key observation that the properties of strict NE suffices to adapt the analysis in [Meng et al., 2023] to zero-sum games (not strongly-convex-strongly-concave games) with a strict NE. ”  We also added a remark before the proof of Theorem 1. Thank you for pointing that out!
>
> > What is the exact difference between "Convergence of the iterates" and "Convergence in the Duality Gap"? What are the advantages of "Convergence of the iterates" compared to "Convergence in the Duality Gap"?
>
> A: Thank you for the question. We clarify the difference between "Convergence of the iterates" and "Convergence in the Duality Gap" below.
>
> Given a sequence of iterates $\{z^t\}$, **convergence of the iterates** means $\lim_{t \rightarrow \infty} z^t$ exists and is a Nash equilibrium; **convergence of the duality gap** means $\lim_{t\rightarrow \infty} \textnormal{DualGap}(z^t)  = 0$. If we only focus on asymptotic convergence, the convergence of the iterates is stronger since it implies convergence of the duality gap; but convergence of the duality gap does not imply convergence of the iterates. One example is that the iterates could cycle and become closer and closer to the set of Nash equilibria (so the duality gap goes to 0), but the trajectory does not converge to one point. It also illustrates an advantage of convergence in the iterate over convergence in the duality gap, as convergence in the iterate guarantees the stability of the actual sequence of iterates played. Convergence in the iterate is also called point convergence and is desirable in the optimization community [1].
>
> > Why does the first author in prior works use tangent residual to claim "last-iterate convergence," while this paper distinguishes between "Convergence of the iterates" and "Convergence in the Duality Gap"?
>
> A: We note that convergence in the iterate is usually asymptotic unless for strongly-convex-strongly-concave problems. However, we could analyze **rate** of convergence (convergence rate) for the convergence of the duality gap. We say an algorithm has a convergence rate $f(T)$ in duality gap if $\textnormal{DualGap}(z^T) \le f(T)$ for all $T$.
> Taking extragradient (EG) as an example, EG is well-known to have convergence in the iterate (see [2]) which also implies convergence in the duality gap. However, its rate of convergence was unclear until [3]. The main contribution of [3] is to provide the convergence rate of EG, showing that $\textnormal{DualGap}(z^T) \le O(1/\sqrt{T})$. In the current paper, for ExRM+ and SPRM+, convergence in the duality gap is relatively standard to prove by classical results. Our main technical contribution is to prove the stronger notion of convergence in the iterate. We leave the convergence rate regarding the duality gap as an interesting open question.
>
>
> > Is the claim in the conclusion about "linear-rate convergence in iterates" correct?
>
> A: Thanks for pointing it out! This phrase is indeed a bit confusing.  We have changed it to "linear last-iterate convergence". It means that restarting versions of ExRM+ and SPRM+ have a linear last-iterate convergence rate in the duality gap $\textnormal{DualGap}(z^T) \le O( (1-c)^T)$.
>
> [Meng et al., 2023] Meng, Linjian, Zhenxing Ge, Wenbin Li, Bo An, and Yang Gao. "Efficient last-iterate convergence algorithms in solving games." arXiv preprint arXiv:2308.11256 (2023).
>
> [1] Sedlmayer, Michael, Dang-Khoa Nguyen, and Radu Ioan Bot. "A fast optimistic method for monotone variational inequalities." International Conference on Machine Learning. PMLR, 2023.
>
> [2] Facchinei, Francisco, and Jong-Shi Pang, eds. "Finite-Dimensional Variational Inequalities and Complementarity Problems." 2003
>
> [3] Cai, Yang, Argyris Oikonomou, and Weiqiang Zheng. "Finite-time last-iterate convergence for learning in multi-player games." Advances in Neural Information Processing Systems 35 (2022): 33904-33919.

---

> > ### Comment · Reviewer_QZ5e · 2024-11-24
> >
> > Thank you for the authors' responses. I have raised my score, but I have one final question. From the perspective of learning an NE, what is the distinction between "Convergence of the iterates" and "Convergence in the Duality Gap"? If our sole objective is to learn an NE, these two types of convergence seem indistinguishable.

---

> > > ### Author Response · Authors · 2024-11-25
> > >
> > > We thank the reviewer for acknowledging our response. Regarding your question, we agree that convergence in the duality gap suffices if one's only goal is to compute a Nash equilibrium. We remark again that from the perspective of learning dynamics, convergence in iterate has the additional desirable property that ensures the stability of the dynamics.

---

### Official Review · Reviewer_yK8z · 2024-11-02

**Soundness:** 3
**Presentation:** 3
**Contribution:** 2
**Rating:** 6
**Confidence:** 3

**Summary:**

The paper studies the last iterate convergence of regret matching and its variant in two-player zero-sum games. The results show that RM+ and its predictive variants may not converge, but the extra gradient and optimistic gradient varients can converge.

**Strengths:**

The last iterate convergence of RM algorithms is important and the nonconvergence results are new and interesting. The paper is well-written and the results are sound.

**Weaknesses:**

The main weakness i that RM+ and its variants are mainly applied in EFGs, and so is the paper introduced and motivated. However, the results in the paper only hold for normal-form games. It is hard to be convinced that the results can be extended to EFGs without major revisions, so the motivation to study RM+ in NFGs is insufficient.

The other major problem is that ExtraRM+ is not no-regret, this somewhat contradicts the purpose of regret-matching.

**Questions:**

1. Can you elaborate on the motivation behind considering RM+ algorithm in NFGs? Why are they important in NFGs, given that there are convergence algorithms like EG/OG.
2. How can the results be extended to EFGs?
3. Is it possible to modify EXRM+ to be no regret? Is SPRM+ no regret? ( i believe OG is no regret, so does this extend to SPRM+?)

---

> ### Author Response · Authors · 2024-11-18
>
> Thank you for your positive review and comments. We address your questions below.
>
> > Can you elaborate on the motivation behind considering RM+ algorithm in NFGs? Why are they important in NFGs, given that there are convergence algorithms like EG/OG.
>
> > How can the results be extended to EFGs?
>
> A: This is a great question! EG/OG are popular algorithms for NFGs, and their last-iterate convergence properties have been extensively studied and are now fairly clear. RM+ algorithms are also popular for solving zero-sum games, especially for EFGs. However, little is known about their last-iterate convergence properties. This motivates our study on RM+ algorithms in NFGs (a special case of EFGs). Moreover,
>
> 1. Our non-convergence counterexample shows that many RM+ variants, including RM+, alternating RM+, PRM+ do not converge in the last-iterate in some simple NFGs, which implies that this non-convergence persists in EFGs.
> 2. Our convergence results for NFGs provide the first step toward understanding last-iterate convergence results in EFGs. The main challenge for establishing last-iterate convergence is the non-monotonicity of the regret operator, which we overcome by using the geometric structure of the limit points of the dynamics (lemma 4 in our paper). We require similar characterizations for EFGs to extend our results. We are hopeful that our techniques could be extended to EFGs, given that it is common to study first the properties of algorithms over NFGs before extending the results to EFGs, e.g. the last-iterate convergence of optimistic OMD was obtained first for NFGs in [1] before being proved in [2] for EFGs.
>
>
> 3. RM+-type algorithms involve non-monotone operators and differ from EG/OG, which concerns monotone operators. Our analysis gives the first convergence in iterate result for an operator with a Minty solution and is of independent interest.
>
>
> >Is it possible to modify EXRM+ to be no regret? Is SPRM+ no regret? ( i believe OG is no regret, so does this extend to SPRM+?)
>
> A: ExRM+ is not no-regret. We study ExRM+ since it has been proposed in previous works and is a natural analog of EG. SPRM+ is indeed no regret. This follows from (1) PRM+is no-regret [3] and (2) SPRM+ is PRM+ where the projection is on the clipped orthant (we chop out the origin).
>
> [1] Chen-Yu Wei, Chung-Wei Lee, Mengxiao Zhang, and Haipeng Luo. Linear last-iterate convergence in constrained saddle-point optimization. In International Conference on Learning Representations (ICLR), 2021.
>
> [2] Chung-Wei Lee, Christian Kroer, and Haipeng Luo. Last-iterate convergence in extensive-form games. Advances in Neural Information Processing Systems, 34:14293–14305, 2021.
>
> [3] Farina, Gabriele, Christian Kroer, and Tuomas Sandholm. "Faster game solving via predictive blackwell approachability: Connecting regret matching and mirror descent." Proceedings of the AAAI Conference on Artificial Intelligence. Vol. 35. No. 6. 2021.

---

### Official Review · Reviewer_SVSD · 2024-11-03

**Soundness:** 3
**Presentation:** 4
**Contribution:** 3
**Rating:** 8
**Confidence:** 4

**Summary:**

Learning algorithms for games is one of the fundamental research areas in algorithmic game theory. In this direction the authors of the paper study the (variants of) Regret Matching+ (RM+) algorithms in terms of last/best-iterate convergence.

Firstly, the authors give instances that (variants of) the RM+ algorithm diverges for last/best-iterate convergence even if the Nash equilibrium of the game is unique, so quasi-strict NE. On the other hand, interestingly enough, if there is a unique Nash equilibrium that is strict, so pure NE, they give a Theorem that PM+ converges in last-iterate, but without giving any rate guarantee.

After this analyzing properties of the operator, they give last-iterate convergence and best-iterate convergence for the EXRM+ algorithm giving also convergence rate for the duality gap matching the state of the art O(1/sqrt{t}) for last-iterate algorithms. Similarly, they prove last/best-iterate convergence for the SPRM+ algorithm with same rate of convergence guarantee. Furthermore, they give, in my opinion a very interesting, variant of them that achieves linear last iterate convergence.

Finally the authors give experiments to give evidence for their results.

**Strengths:**

In my opinion I think that the problem of learning algorithms in games is one of the fundamental problems in algorithmic game theory. Thus, the contribution of this paper is significant since not only the authors give results for variants of the RM+ algorithm (a well-known algorithm), but also give nice and novel techniques and directions, such as the algorithms with the restart, that can potentially help the research to go further on the learning algorithms. All these along with the quality of the paper make it a nice piece of research work.

**Weaknesses:**

My only minor concern is that in Theorem 4 and 7 I think that η must be constant (not function of t) since β must be upper bounded by a constant strictly less than 1 in order to have linear (geometric/inverse exponential) convergence. Thus, I think, if I am correct, that this should be explicitly stated in the Theorems.

**Questions:**

Do you have any insight about the rate of convergence of the result of the Theorem 1? I think this will be a good direction for future work.

---

> ### Author Response · Authors · 2024-11-18
>
> Thank you for the very positive review. We address your questions below.
>
> > My only minor concern is that in Theorem 4 and 7 I think that $\eta$ must be constant (not function of t) since $\beta$ must be upper bounded by a constant strictly less than 1 in order to have linear (geometric/inverse exponential) convergence. Thus, I think, if I am correct, that this should be explicitly stated in the Theorems.
>
> A: In algorithms 5 and 6, we set the step size $\eta$ to be constant (independent of $t$). In the updated version, we have restated it (we added ''with constant step size $\eta$") in the theorems for clarity.
>
>
> >Do you have any insight about the rate of convergence of the result of the Theorem 1? I think this will be a good direction for future work.
>
> A: In fact, as we remark at the end of the proof of Theorem 1 in the appendix (page 16, Line 862), our proof gives that RM+ has a $O(1/\sqrt{T})$ best-iterate convergence rate. This rate depends on the constant induced by the strict NE property, which may be very small. We agree that investigating the problem-independent best-iterate convergence rate and last-iterate convergence rate of RM+ in games with strict NE is a very interesting future direction.

---

> ### Comment · Reviewer_SVSD · 2024-11-26
> **Response to the Authors**
>
> I would like to thank Authors for their answer. My questions are completely answered and I will keep my initial score.

---

### Official Review · Reviewer_dd2U · 2024-11-04

**Soundness:** 3
**Presentation:** 3
**Contribution:** 3
**Rating:** 6
**Confidence:** 3

**Summary:**

The authors study last-iterate convergence  for Regret Matching + (RM+) algorithm and its recent variants PRM+ (Predictive RM+), ExRM+ (Extragradient RM+) and Smooth Predictive RM+(SPRM+) in two-player zero-sum bimatrix games. They show asymptotic last-iterate convergence of these variants (linear last-iterate with restarts). They also show ($1/\sqrt{t})$ best-iterate convergence.

**Strengths:**

1) The paper is generally well written .

2) They show that RM+ and PRM+ have last-iterate convergence (asymptotically) only in the case of having a strict-NE and provide a numerical example that this assumption is necessary.

3) Moving on to the smooth variants of RM+ introduced in [Farina et al., 2023], they show that the operators corresponding to these algorithms are non-monotone and satisfy only a weak condition known as the Minty condition.

4) Furthermore, they provide structural analysis on the limit points of the dynamics and show last-iterate convergence and this can be extended with additional regularity assumptions (metric-subregularity) and with restarts to show linear last-iterate convergence as well.

References:
Farina, Gabriele, et al. "Regret matching+:(in) stability and fast convergence in games." Advances in Neural Information Processing Systems 36 (2024).

**Weaknesses:**

1) Their paper is restricted to two-player zero-sum games (bimatrix) games. It would be great to get a clarity as to if these techniques can be extended for general convex-concave games? (And also for general convex sets).

2) Is it possible to obtain asymptotic rates of convergence for the last-iterate analysis shown here?

**Questions:**

See Weaknesses.

---

> ### Author Response · Authors · 2024-11-18
>
> Thank you for the very positive review. We address your questions below.
>
> > Their paper is restricted to two-player zero-sum games (bimatrix) games. It would be great to get a clarity as to if these techniques can be extended for general convex-concave games? (And also for general convex sets).
>
> A: We would like to clarify that the regret matching-type algorithms that we study in our paper are specifically designed for regret minimization over simplices. As such, studying general convex sets would require different algorithms. It is possible to extend Blackwell approachability in this way, see e.g. [1,2]. Extending our results to this case is an interesting future direction, but most likely a general analysis would be very hard.
>
> Extending our results to convex-concave problems over simplexes does not have this issue: in this case, one can take subgradients to perform regret minimization via regret matching. The results may generalize if subgradients are bounded, but we are not sure whether more conditions are needed for last-iterate convergence.
>
> >Is it possible to obtain asymptotic rates of convergence for the last-iterate analysis shown here?
>
> A: We assume that your question is about getting a specific rate of convergence, as opposed to the asymptotic rate-free guarantee. If you had something else in mind, please feel free to ask us to clarify.
>
> Our results only give a rate-free asymptotic last-iterate convergence for RM+ (when strict NE exists), ExRM+, and SPRM+. We also prove concrete rates for the best-iterate convergence of  ExRM+ and SPRM+, and linear last-iterate convergence rates for their restarting variants. We acknowledge that we do not have last-iterate convergence rate results for ExRM+ and SPRM+. Proving such results usually requires the operator to be monotone and a sophisticated potential function argument (like the proof for the Extra-Gradient and Optimistic-Gradient methods). These techniques have not been generalized to operators satisfying only the Minty condition, so progress on this front would likely be a challenging result of independent interest. We leave obtaining last-iterate convergence rates for ExRM+ and SPRM+ is an interesting future direction.
>
>
> [1] Grand-Clément, Julien, and Christian Kroer. "Conic Blackwell algorithm: Parameter-free convex-concave saddle-point solving." Advances in Neural Information Processing Systems 34 (2021): 9587-9599.
>
> [2] D'Orazio, R., & Huang, R.. Optimistic and adaptive Lagrangian hedging. AAAI 2021.

---

> > ### Comment · Reviewer_dd2U · 2024-11-25
> > **Small Clarification.**
> >
> > I thank the authors for their responses and clarification. In regards to the last-iterate rates, I was wondering, if there is a possibility of showing _local_ convergence rates, as studied here : (https://arxiv.org/abs/1807.04252), for example. Please note that this is only for curiosity sake and does not affect my existing positive viewpoint of the work.

---

> > > ### Author Response · Authors · 2024-12-02
> > >
> > > Thank you for the question! We note that Theorem 3.3 in [1] only gives local convergence *without a rate*: it proves that if the iterates enter a neighborhood of the unique Nash equilibrium, then the OMWU dynamics converge to the Nash equilibrium in the limit. We believe analyzing local convergence rates of ExRM+, SPRM+, and their variants requires new insights and is interesting future work.
> > >
> > > [1] Daskalakis, Constantinos, and Ioannis Panageas. "Last-Iterate Convergence: Zero-Sum Games and Constrained Min-Max Optimization." 10th Innovations in Theoretical Computer Science (2019). https://arxiv.org/abs/1807.04252

---

### Meta-Review · Area_Chair_AFu3 · 2024-12-21

**Metareview:**

A great paper in last-iterate convergence of learning in games!

**Additional Comments On Reviewer Discussion:**

NA

---

### Decision · Program_Chairs · 2025-01-22

Accept (Poster)